EMBO
reports

# DSK2-mediated degradation of F-box protein LAO1 and class I TCPs modulates the nitrogen starvation response

Yuanyuan Li[1], Shuyang Cheng [1], Xu Jin[1], Ruoxuan Wu[1], Yiyi Guo [1], Dezhi Wu[2,3] & Jie Dong [1✉]

## Abstract

Plants have evolved intricate strategies to cope with various abiotic stresses. Ubiquitin-mediated protein degradation plays a key role in plant development as well as abiotic stress tolerance. In this study, we identify LAO1, an F-box protein with unknown function, as a negative regulator of plant fitness during nitrogen starvation. DOMINANT SUPPRESSOR OF KAR 2 (DSK2) interacts with and mediates the autophagic degradation of LAO1 protein during nitrogen starvation. The loss of *LAO1* improves the fitness of an autophagy-deficient mutant, *atg5-1*, under nitrogen starvation. Intriguingly, mutations in *DSK2* facilitate rather than reduce plant growth after nitrogen starvation. This unexpected effect of *DSK2* knockout led us to discover that DSK2 also interacts with and degrades a group of class I TCP transcription factors. Phenotypic observations demonstrate that class I TCPs are crucial for plant adaptation to nitrogen starvation. Moreover, genetic analyses indicate that class I TCPs function downstream of LAO1 and counteract its negative effects. Collectively, our findings unveil a previously undescribed regulatory network governing plant fitness during nitrogen starvation.

**Keywords** *Arabidopsis thaliana*; F-box Protein; Nitrogen Starvation; DSK2; Class I TCP Transcription Factors
**Subject Categories** Autophagy & Cell Death; Plant Biology; Post-translational Modifications & Proteolysis

## Introduction

As sessile organisms, plants cannot relocate to escape deleterious abiotic environmental stresses. Challenges such as nitrogen starvation significantly impact plant growth, development, and crop yield (Gong et al, 2020; Sanagi et al, 2021; Yang et al, 2022; Zhang et al, 2021). To adapt, plants have evolved complex regulatory networks encompassing transcriptional reprogramming, translational optimization, and post-translational regulation. These

mechanisms facilitate the sensing of stress signals into developmental responses (Zhang et al, 2022). Consequently, plants can modulate their growth in accordance with each stress, optimizing survival and reproductive success.

Ubiquitin-mediated protein degradation is pivotal in plant response to abiotic stresses. Polyubiquitinated proteins may be degraded through either the autophagy or the proteasome pathway (Floyd et al, 2012; Kraft et al, 2010). Autophagy, a process requiring the coordinated function of multiple *AUTOPHAGY-RELATED* (*ATG*) genes, is a conserved mechanism critical for plant abiotic stress responses and nutrient recycling during nutrient deficiency (Avin-Wittenberg, 2019; Marshall and Vierstra, 2018). Upon starvation or stress conditions, autophagy is promptly activated via a tightly controlled signaling cascade involving the assembly of ATG1 kinase complex, lipids delivery to phagophore, and phagophore decoration with PI3P and the central component ATG8 (Ohsumi 2001; Pu et al, 2017; Suttangkakul et al, 2011; Xiong et al, 2005; Yoshimoto et al, 2004). This process culminates in the initiation of three principal types of autophagy: micro-autophagy, macroautophagy, and mega-autophagy, enabling plants to cope with nutrient-deficient conditions (van Doorn and Papini, 2013). In addition, lipidated ATG8 can selectively interact with various receptor proteins, facilitating the selective autophagic degradation of specific target proteins or organelles. One such receptor is DOMINANT SUPPRESSOR OF KAR 2 (DSK2), encoded by two paralogs, DSK2A and DSK2B, in *Arabidopsis* (Farmer et al, 2010). These paralogs share 87% amino acid sequence identity. DSK2 binds to polyubiquitinated proteins via its C-terminal ubiquitin-associated (UBA) domain and targets them for degradation (Lin et al, 2011).

The ubiquitin-proteasome system (UPS) is a critical pathway for targeted protein degradation, ensuring protein homeostasis in vivo. It governs a variety of plant developmental processes, including hormone signaling, photomorphogenesis, circadian clock regulation, embryogenesis, and abiotic stress response (Vierstra, 2009; Xu and Xue, 2019). In *Arabidopsis*, the F-box proteins, comprising over 700 members, represent the largest E3 ubiquitin ligase family (Gagne et al, 2002). It has been challenging to investigate their functions due to functional redundancy among family members. We previously reported that the assembly of SCF complex are regulated by both substrate phosphorylation and

[1]Institute of Crop Science, College of Agriculture and Biotechnology, Zhejiang University, Hangzhou 310058, China. [2]College of Agronomy, Hunan Agricultural University, Changsha 410128, China. [3]Yuelushan Laboratory, Changsha 410128, China. ✉E-mail: jie_dong@zju.edu.cn

 

COP9 signalosome (CSN) complex, which may be important for the activity of certain F-box proteins (Dong et al, 2017; Dong et al, 2024). Although the role of F-box protein family in phytohormone signaling, photomorphogenesis and flowering has been extensively studied (Lechner et al, 2006), the detailed physiological function of SCF complex and its interplay with other signals in plant abiotic stress responses are yet to be resolved.

*Arabidopsis* harbors 24 TEOSINTE BRANCHED1, CYCLOIDEA, PROLIFERATING CELL FACTOR 1 and 2 (TCP) transcription factors, 13 of which belong to class I TCPs (Navaud et al, 2007; Martín-Trillo and Cubas, 2010). Class I TCPs orchestrate diverse processes, including germination, hypocotyl elongation, cell proliferation and elongation, endoreplication, circadian clock, flowering, and nitrate signaling (Zhang et al, 2019b; Spears et al, 2022; Xu et al, 2020; Resentini et al, 2015; Ferrero et al, 2019; Peng et al, 2015; Zhang et al, 2018; Pruneda-Paz et al, 2009). TCP14 and TCP15, the activity of which are repressed by DELLA proteins, are required for GA-induced seed germination (Xu et al, 2020; Resentini et al, 2015). TCP15 facilitates PIF4-induced hypocotyl elongation under ambient temperature (Ferrero et al, 2019). TCP7, possibly together with TCP8, TCP14, TCP15, TCP21, TCP22, and TCP23, activates *CYCD1;1* expression to regulate endoreplication (Peng et al, 2015; Zhang et al, 2018). TCP21/CHE is a key regulator of circadian clock (Pruneda-Paz et al, 2009). TCP7 and TCP15 activate *SOC1* expression to promote flowering (Lucero et al, 2017; Li et al, 2021). Under nitrate starvation, TCP20 interacts with NLP6 and NLP7, two NIN-like transcription factors, to regulate genes involved in root development and root system architecture (Guan et al, 2017; Guan et al, 2014).

In this study, we uncover a previously undescribed molecular framework involving the conserved ubiquitin receptor protein DSK2, the F-box protein LAO1, and the class I TCP transcription factors which collectively regulate plant adaptation to nitrogen starvation. Specifically, DSK2 targets LAO1 for autophagic degradation to enhance plant fitness during nitrogen starvation, while DSK2-mediated TCP degradation exerts an opposing effect. Genetically, class I TCP transcription factors function downstream of LAO1 to modulate plant response to nitrogen starvation. Overall, our results provide insights into the intermolecular interactions that underpin plant adaptation to nitrogen starvation.

## Results

### The F-box protein LAO1 is a negative regulator of plant nitrogen starvation tolerance

In a previous study, we performed an immunoprecipitation coupled with mass spectrometry (IP-MS) using CUL1 as bait in both Col and *csn1-10* mutant backgrounds, identifying a group of F-box proteins whose association with CUL1 is regulated by the CSN complex and CUL1 neddylation (Dong et al, 2024). AT1G67480 is one of the five F-box proteins that showed increased binding with CUL1 in *csn1-10* mutant background, but its physiological function is unknown. To elucidate its role in the regulation of plant development, we generated two independent overexpression lines of AT1G67480 (Appendix Fig. S1). Both lines displayed an accelerated leaf senescence phenotype when compared to the wild type Columbia-0 (hereafter referred to as Col; Fig. 1A).

Furthermore, we isolated a T-DNA mutant of AT1G67480 (Fig. 1B). We observed that the leaves of this mutant stayed green longer than those of Col (Fig. 1C), indicating that AT1G67480 encodes a factor promoting plant leaf senescence. Hence, we refer to AT1G67480 as *LAO1* (the Chinese pronunciation of 'senescence').

Precocious leaf senescence can result from multiple developmental and environmental factors, including nutrient supply and utilization deficiencies (Schulze et al, 1994). To unravel the possible role of *LAO1* in planta, we analyzed its expression profile using two public *Arabidopsis* RNA-seq databases. The *Arabidopsis* eFP browser (https://bar.utoronto.ca/eplant/) indicated active *LAO1* expression in rosette and cauline leaves (Winter et al, 2007; Fig. EV1A). The plant public RNA-seq database (https://plantrnadb.com/athrdb/), which comprises more than 20,000 public *Arabidopsis* RNA-seq libraries, showed that *LAO1* expression is responsive to various abiotic stresses, including water stress, wounding, dark, and nutrient deficiency (Yu et al, 2022; Fig. EV1B). More specifically, *LAO1* was markedly downregulated in databases associated with nitrogen-limited treatments, while nitrogen-rich treatments enhanced its transcription (Fig. EV1C; Alvarez et al, 2020; Soto-Cardinault et al, 2023; Zhang et al, 2019a; Rallapalli et al, 2014; Crawford et al, 2020; Drechsel et al, 2013; Zhang et al, 2020a; Denyer et al, 2019; Narsai et al, 2017; Sura et al, 2017; Rasheed et al, 2016; Bhandari et al, 2019; Rawat et al, 2015; Xiao et al, 2023; Kannan et al, 2018; Liu et al, 2014; Ishihara et al, 2019; Zhang et al, 2020b; Brooks et al, 2019; Awazu et al, 2018; Survila et al, 2016; Hou and Ma 2017; Schroeder et al, 2019; Ré et al, 2019; Dataset EV1). In addition, only nitrogen deficiency, but not other nutrient deficiencies, decreased *LAO1* transcript levels (Fig. EV1D; Nishida et al, 2017; Dataset EV1), and this correlation between nitrogen availability and *LAO1* expression was consistent across a variety of published databases (Fig. EV1E; Gaudinier et al, 2018; Li et al, 2023; Takami et al, 2018; Thieme et al, 2015; Dataset EV1). These observations indicate a tight association between *LAO1* expression levels and nitrogen availability.

The molecular analyses above prompted us to hypothesize that *LAO1* may play a role in the regulation of plant adaptation to fluctuating nitrogen conditions. To validate this hypothesis, we grew Col, *LAO1* OE1, *LAO1* OE2, and *lao1* T-DNA mutant (*lao1*(T)) seedlings on 1/2 MS plates for 4 days, and then transferred them to nitrogen-free 1/2 MS (1/2 MS-N) plates for nitrogen starvation treatment, followed by recovery on 1/2 MS plates before chlorophyll contents and biomass were measured. *lao1*(T) mutant showed significantly increased chlorophyll content and biomass compared to Col (Fig. 1D,E). Conversely, *LAO1* OE1 and OE2 seedlings exhibited a marked reduction in chlorophyll content and biomass compared to Col after nitrogen starvation treatment (Fig. 1F,G). Collectively, these findings suggest that LAO1 exerts a negative influence on *Arabidopsis* seedling fitness under nitrogen-deficient conditions.

### Nitrogen deficiency downregulates *LAO1* mRNA and protein abundance

We further asked whether nitrogen availability could regulate LAO1 abundance. A time-course reverse transcription-quantitative polymerase chain reaction (RT-qPCR) analyses revealed that 12 h of nitrogen starvation significantly downregulated *LAO1* expression, and this downregulation persisted up to 96 h (Fig. 2A).

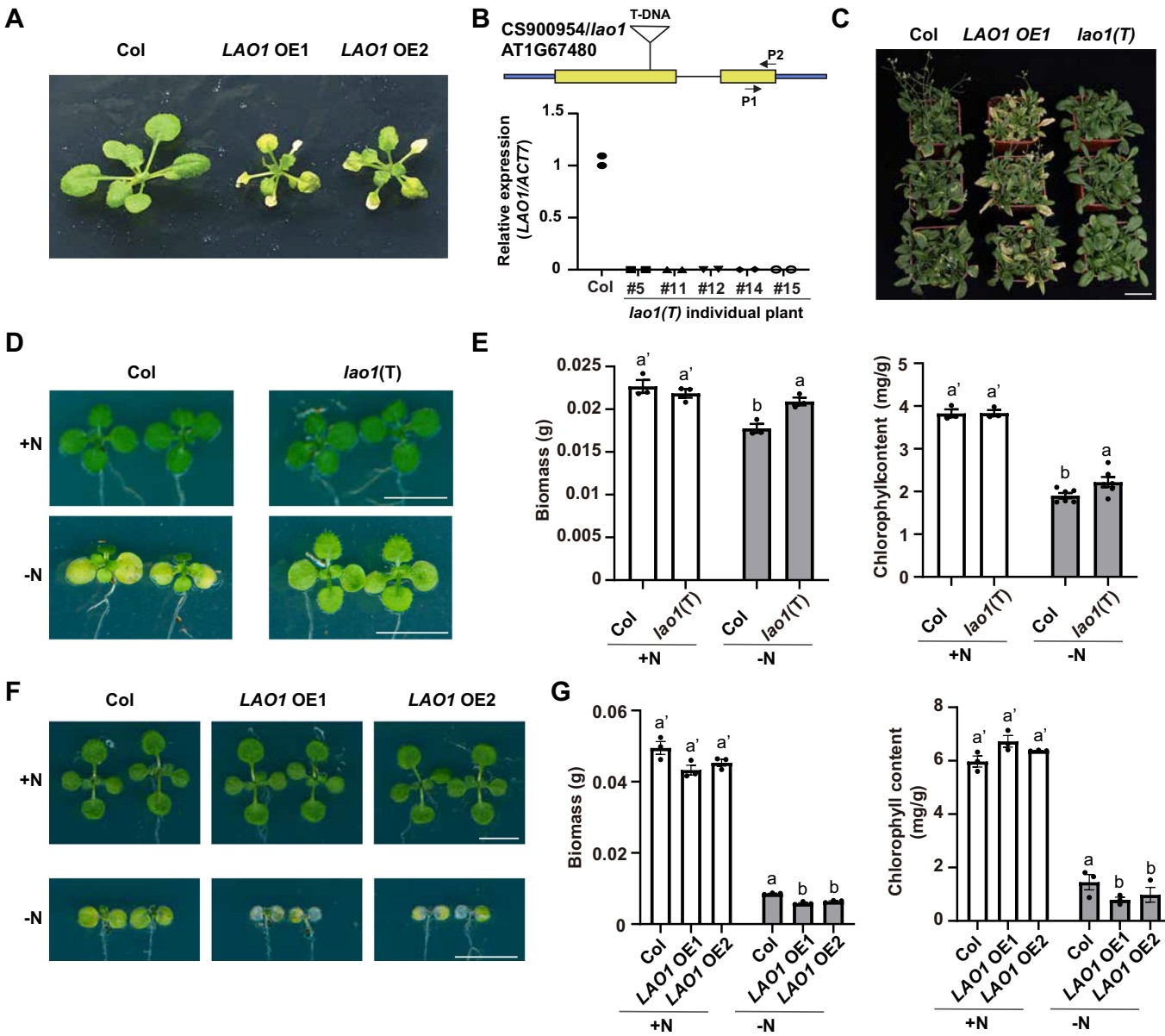

**Figure 1. The F-box protein LAO1 is a negative regulator of plant nitrogen starvation fitness.**

(A) Images of 25-day-old Col, *35S: HA-LAO1* #1 (*LAO1* OE1), and *35S: HA-LAO1* #2 (*LAO1* OE2) grown under a long-day condition (16 h light/ 8 h dark) at 22 °C in a growth chamber. (B) Isolation of a T-DNA mutant of *LAO1*, *lao1*(T), and the confirmation of *lao1*(T) mutant by RT-qPCR. Expression levels of *LAO1* were shown as individual datapoints of technical replicates ($n = 2$). (C) Images of 4-week-old Col, *LAO1* OE1, and *lao1*(T) mutant grown under a long-day condition (16 h light/8 h dark) at 22 °C in a growth chamber. Scale bar: 5 cm. (D, E) *lao1*(T) mutant showed significantly increased chlorophyll content and biomass compared to Col after nitrogen starvation treatment. Four-day-old seedlings were treated with nitrogen starvation for 6 days followed by 4 days of recovery. (D) A representative image. Scale bars: 0.5 cm. (E) Determination of the biomass and chlorophyll content of Col and *lao1*(T) mutant seedlings. (F, G) *LAO1* OE seedlings exhibited significantly decreased chlorophyll content and biomass compared to Col after nitrogen starvation treatment. Four-day-old seedlings were treated with nitrogen starvation for 4 days followed by 2 days of recovery. (F) A representative image. Scale bars: 0.5 cm. (G) Determination of the biomass and chlorophyll content of Col and *LAO1* OE (*LAO1* OE1 and *LAO1* OE2) seedlings. Data were shown in mean ± SEM of at least 3 biological replicates. Each biological replicate contained at least 15 seedlings. In (E) and (G), all the statistical significances were calculated using one-way ANOVA ($p < 0.05$, ANOVA followed by Tukey's post hoc comparison test) analysis. Different letters denoted significant differences. Source data are available online for this figure.

To further support this observation, we generated transgenic plants expressing the β-glucuronidase (GUS) under the control of *LAO1* promoters (*proLAO1(3k): GUS* and *proLAO1(2k): GUS*). We treated 7-day-old *proLAO1(3k): GUS* #1 & #3 and *proLAO1(2k): GUS* #3 & #8 transgenic seedlings grown on 1/2 MS plates with 72 h of

nitrogen starvation. The GUS signals in all the transgenic lines were significantly reduced after nitrogen starvation (Fig. 2B), confirming that nitrogen starvation inhibits *LAO1* transcription.

To assess the impact of nitrogen starvation on the protein levels of LAO1, we monitored HA-LAO1 protein levels after transferring

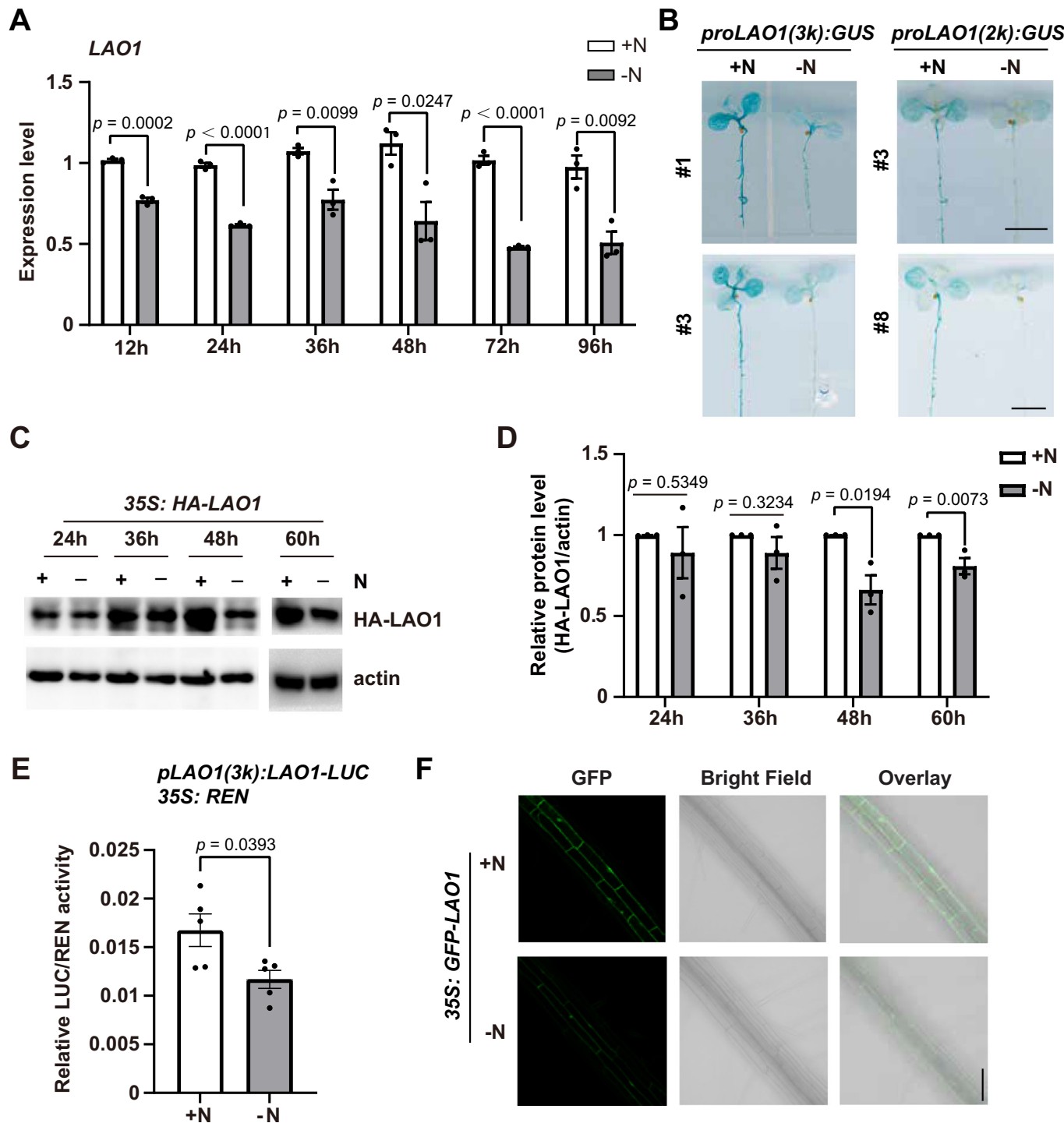

**A** LAO1

**B** *proLAO1(3k):GUS* *proLAO1(2k):GUS*

**C** *35S: HA-LAO1*

**D**

**E** *pLAO1(3k):LAO1-LUC 35S: REN*

**F** GFP | Bright Field | Overlay — *35S: GFP-LAO1*

*35S: HA-LAO1* seedlings, initially grown on 1/2 MS plates, to nitrogen-free 1/2 MS plates (1/2 MS-N) for nitrogen starvation treatment. As this is a long-term treatment, to rule out other factors that potentially impact LAO1 protein abundance, seedlings transferred to normal 1/2 MS plates for the same duration served as the control. We further compared the HA-LAO1 protein levels under nitrogen starvation conditions to those under normal nitrogen conditions at each time point. Immunoblot analyses showed that the HA-LAO1 protein levels at 48 and 60 h of nitrogen starvation were significantly lower than those at the corresponding time points under normal nitrogen conditions (Fig. 2C,D), suggesting that nitrogen starvation may influence HA-LAO1 protein abundance. Additionally, we blocked proteins synthesis with cycloheximide (CHX), and monitored the turnover of HA-LAO1 protein under both normal nitrogen and nitrogen starvation conditions. We observed that nitrogen starvation significantly

**Figure 2. Nitrogen deficiency downregulates *LAO1* mRNA and protein abundance.**

(A) The relative expression levels of *LAO1* under normal nitrogen (+N) or nitrogen starvation (-N) conditions. Four-day-old Col seedlings grown on 1/2 MS plates (+N) were transferred to 1/2 MS-N plates (-N) for indicated time, and total RNA was extracted for RT-qPCR analyses. *Actin 7* (*ACT7*) was used as internal control. Relative expression levels of *LAO1* were shown in mean ± SEM of 3 independent experiments. (B) GUS histochemical staining of transgenic *Arabidopsis* expressing GUS driven by *LAO1* promoter. Seven-day-old *proLAO1(3k)*: GUS #1& #3 and *proLAO1(2k)*: GUS #3& #8 seedlings grown on 1/2 MS plates (+N) were transferred to 1/2 MS-N plates (-N) for 72 h followed by GUS staining process. Scale bars: 0.5 cm. (C, D) Analyses of HA-LAO1 protein levels after nitrogen-starvation treatment. Seven-day-old *35S: HA-LAO1* seedlings grown on 1/2 MS plates (+N) were treated with nitrogen starvation for indicated time, and total proteins were extracted. Western blots were performed using anti-HA and anti-actin antibodies. (C) A representative western blot result. (D) Relative protein levels of HA-LAO1 shown in mean ± SEM of 3 independent experiments. (E) The luminescence ratio of LAO1-LUC/REN under nitrogen starvation (-N) and normal nitrogen (+N) conditions. The coding sequencing of firefly luciferase (LUC) was fused with that of LAO1 and driven by *LAO1* native promoter. Rennilla LUC (REN), the internal control, was driven by 35S promoter. Transfected protoplasts were either incubated in medium without nitrogen (-N) or with nitrogen (+N). Relative LAO1-LUC/REN activity was shown in mean ± SEM of 5 independent experiments. (F) Confocal imaging of *35S: GFP-LAO1* transgenic plants after nitrogen starvation. Seven-day-old seedlings were treated with nitrogen starvation for 48 h, and the GFP signal in hypocotyls were imaged with confocal microscopy. Scale bar: 15 μm. All the statistical significances were calculated by the Student's *t* test. Source data are available online for this figure.

accelerated the turnover of HA-LAO1 protein (Appendix Fig. S2), indicating that nitrogen starvation destabilizes HA-LAO1 protein in vivo. To corroborate these observations, we measured the relative luminescence of firefly luciferase (LUC), fused with *LAO1* coding sequence and driven by its native promoter, to Rennilla LUC (REN), which is driven by 35S promoter. Dual-luciferase reporter assay suggested that the luminescence ratio of LAO1-LUC/REN under nitrogen starvation condition (-N) was significantly lower than nitrogen-rich (+N) conditions (Fig. 2E). Moreover, we generated *35S: GFP-LAO1* transgenic plants expressing LAO1 fused with green fluorescent protein (GFP) at the N-terminus. Confocal imaging showed that the GFP fluorescence was prominent under normal nitrogen condition but significantly reduced following nitrogen starvation (Fig. 2F). Taken together, nitrogen starvation downregulates LAO1 abundance at both transcriptional and post-translational levels.

## LAO1 abundance is associated with developmental processes involving nitrogen assimilation or remobilization

The observations above promoted us to ask whether LAO1 abundance is associated with in vivo nitrogen level dynamics. Nitrogen is remobilized from older to younger leaves during the senescence of adult plants, while seedling assimilate nitrogen during the development of true leaves (Diaz et al, 2008). During these two processes, we monitored both endogenous *LAO1* transcript level in Col and HA-LAO1 protein level in *35S: HA-LAO1*. We numbered the leaves of 3-week-old adult plants (Fig. EV2A) from group #1 (representing younger leaves) to group #9 (representing older leaves). RT-qPCR analysis of Col adult plants revealed a slight decrease in *LAO1* expression in groups #8 and #9 compared to other leaves (Fig. EV2B). Immunoblot analysis detected a gradual decrease of HA-LAO1 protein levels as leaves getting older (Fig. EV2C). During seedling true leaf development, from 4 to 9 days after stratification (Fig. EV2D), *LAO1* mRNA level remained unchanged (Fig. EV2E), while HA-LAO1 protein levels significantly increased (Fig. EV2F). These results suggest that the abundance of LAO1, particularly at the protein level, is associated with nitrogen levels during plant development in vivo.

To further support this notion, we introduced mutations in *NITRATE REDUCTASE 2* (*NIA2*), encoding a key enzyme for nitrogen assimilation (Wilkinson and Crawford 1991; Wilkinson

and Crawford 1993), via CRISPR-Cas9 mutagenesis technology in both Col and *35S: HA-LAO1* transgenic background (Fig. EV2G,I). RT-qPCR analyses indicated that *LAO1* mRNA level in *nia2 cr* mutant seedlings was comparable to that in Col seedlings (Fig. EV2H). However, the protein level of HA-LAO1 was significantly decreased following the knockout of *NIA2* gene (Fig. EV2J). Taken together, our results demonstrate that local nitrogen availability fine tunes LAO1 abundance in planta, particularly at the level of protein regulation, which relies on normal nitrogen assimilation process.

## Nitrogen deficiency destabilizes LAO1 via autophagy pathway

The results above demonstrate that LAO1 is a crucial factor in plant fitness to nitrogen-starved conditions, and nitrogen level dynamics significantly influence LAO1 protein levels. It is well-established that the autophagy pathway is activated in response to nitrogen starvation (Marshall and Vierstra, 2018). To elucidate the relationship between nitrogen starvation-induced LAO1 protein degradation and autophagy pathway, we treated *35S: HA-LAO1* transgenic plants with liquid 1/2 MS-N containing either CHX alone or CHX combined with E64d (a potent inhibitor of the autophagy pathway). We observed that E64d treatment could inhibit nitrogen starvation-induced fast turnover of HA-LAO1 protein (Fig. 3A), suggesting that autophagy may mediate LAO1 degradation following nitrogen starvation. To provide further evidence, we crossed *35S: HA-LAO1* with autophagy-deficient mutants, *atg5-1* and *atg7-3* (Nolan et al, 2017), generating *35S: HA-LAO1/atg5-1* and *35S: HA-LAO1/atg7-3*, respectively. Immunoblot analyses revealed that nitrogen starvation-induced rapid HA-LAO1 protein degradation was attenuated in both *atg5-1* and *atg7-3* mutant backgrounds (Fig. 3B; Appendix Fig. S3A). To rule out the potential effect of CHX on autophagy activity (Watanabe-Asano et al, 2014), we examined the HA-LAO1 protein abundance in *35S: HA-LAO1*, *35S: HA-LAO1/ atg5-1*, and *35S: HA-LAO1/atg7-3* after 12 and 24 h of nitrogen deficiency treatment in the absence of CHX. Similarly, we observed that nitrogen starvation induced fast degradation of HA-LAO1 in Col background, whereas the degradation is significantly slower in both *atg5-1* and atg*7-3* mutant backgrounds (Fig. 3C; Appendix Fig. S3B). To further verify this observation, we compared the luminescence ratio of LAO1-LUC, driven by *LAO1* native promoter, to REN, in protoplast isolated from Col, *atg5-1*, and

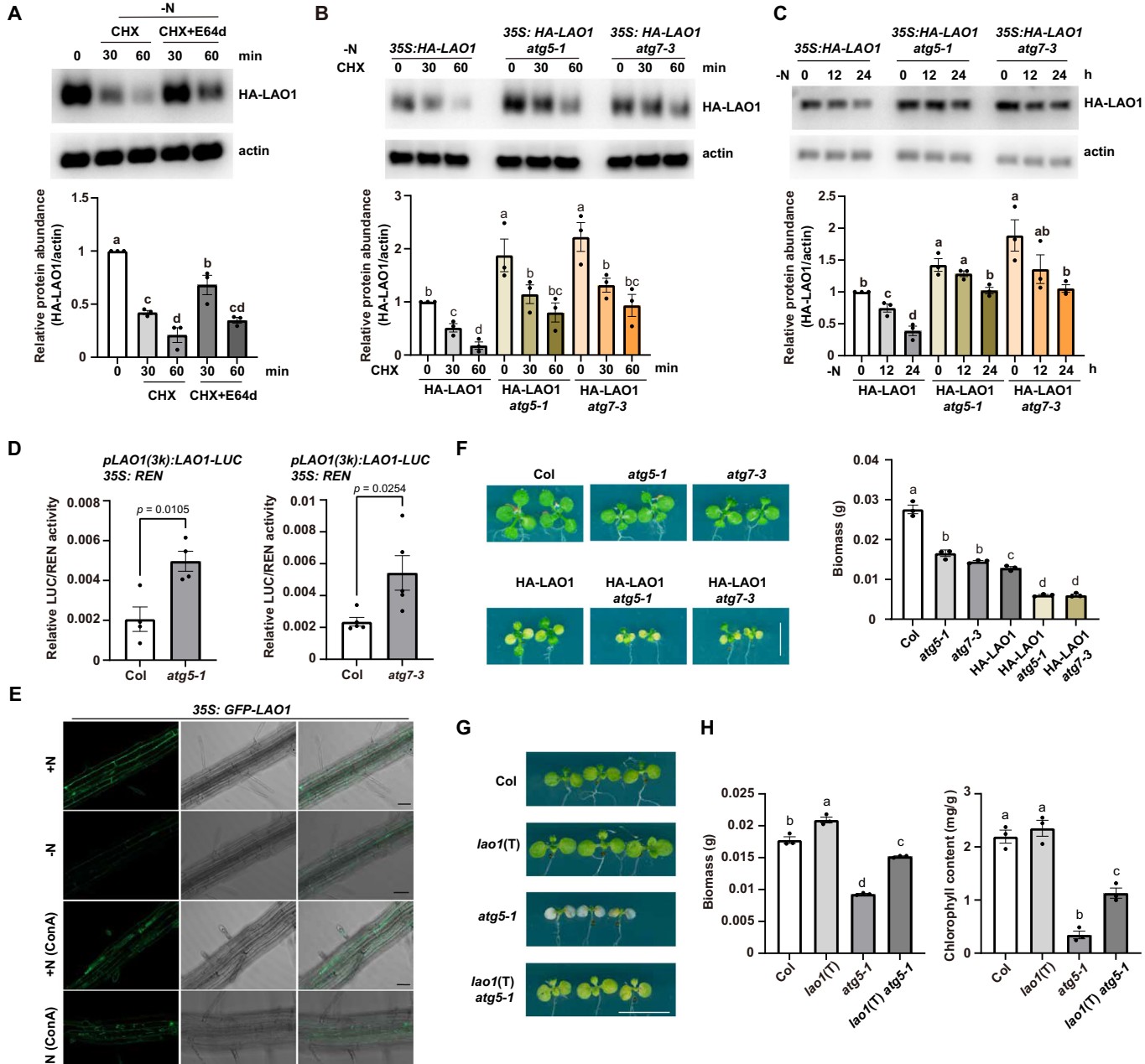

*atg7-3* mutant leaves. Dual-luciferase reporter assay suggested that LAO1-LUC/REN in Col was significantly lower than *atg5-1* and *atg7-3* mutant backgrounds (Fig. 3D). Furthermore, we observed that the GFP signals in *35S: GFP-LAO1* seedlings accumulated in the vacuole under nitrogen starvation conditions in the presence of Concanamycin A (ConA), a V-ATPase inhibitor (Fig. 3E; Zhang et al, 2020c). Taken together, these findings suggest that LAO1 is targeted for autophagic degradation following nitrogen starvation.

To investigate the significance of autophagy-mediated LAO1 protein degradation, we monitored the phenotypes of *35S: HA-LAO1/atg5-1* and *35S: HA-LAO1/atg7-3* after nitrogen starvation. The biomass of *35S: HA-LAO1/atg5-1* and *35S: HA-LAO1/atg7-3* was significantly lower than that of *35S: HA-LAO1* (Fig. 3F), which is consistent with the observation that HA-LAO1 protein levels

were significantly higher in both *atg5-1* and *atg7-3* mutant backgrounds (Fig. 3B,C; Appendix Fig. S3A,B). Furthermore, we generated *lao1*(T) *atg5-1* double mutant by genetic crossing. We observed that mutation in *LAO1* suppressed the phenotypes of *atg5-1* mutant after nitrogen starvation (Fig. 3G,H). Collectively, these findings indicate that LAO1 operates downstream of the autophagy pathway to modulate plant adaptation to nitrogen starvation conditions.

## DSK2 interacts with and mediates protein degradation of LAO1 during nitrogen starvation

Autophagy pathway removes ubiquitinated targets (Marshall and Vierstra, 2018), which prompted us to examine whether nitrogen

◀

**Figure 3.  LAO1 protein undergoes autophagic degradation during nitrogen starvation.**

(A) E64d (a potent inhibitor of the autophagy pathway) inhibited HA-LAO1 protein degradation under nitrogen starvation condition. Seven-day-old *35S: HA-LAO1* seedlings were treated with liquid 1/2 MS-N containing 50 μM cycloheximide (CHX, a protein biosynthesis inhibitor) only (CHX) or 50 μM CHX and 50 μM E64d (CHX+E64d), and total proteins were extracted. Western blots were performed using anti-HA and anti-actin antibodies. Upper, a representative western blot. Lower, relative protein abundance of HA-LAO1 as shown in mean ± SEM of 3 independent experiments. (B, C) HA-LAO1 proteins were more stable in *atg5-1* and *atg7-3* mutant background. (B) Seven-day-old *35S: HA-LAO1, 35S: HA-LAO1/atg5-1* and *35S: HA-LAO1/atg7-3* seedlings were treated with liquid 1/2 MS-N containing 50 μM CHX for indicated time, and total proteins were extracted. (C) Seven-day-old *35S: HA-LAO1, 35S: HA-LAO1/atg5-1* and *35S: HA-LAO1/atg7-3* seedlings grown on 1/2 MS plates (+N) were treated with nitrogen starvation for indicated time, and total proteins were extracted. Western blots were performed using anti-HA and anti-actin antibodies. Numbers indicate relative HA-LAO1 protein levels to the first lane. Upper, a representative western blot. Lower, relative protein abundance of HA-LAO1 as shown in mean ± SEM of 3 independent experiments. (D) The luminescence ratio of LAO1-LUC/REN in Col and *atg* mutant protoplasts. Constructs used for protoplast transfection was the same as Fig. 2E. Relative LAO1-LUC/REN activity was shown in mean ± SEM of 4 independent experiments. All the statistical significances were calculated by the Student's *t* test. (E) LAO1 protein was transported into the vacuole following nitrogen starvation. Seven-day-old seedlings were pretreated with normal 1/2 MS (+N) or nitrogen-free 1/2 MS (-N) medium for 48 h, followed by treatment with 1 μM ConA for 3 h. Scale bars: 50 μm. (F) Enhanced nitrogen starvation sensitivity of *35S: HA-LAO1/atg5-1* and *35S: HA-LAO1/atg7-3* seedlings. Four-day-old seedlings were treated with nitrogen starvation for 24 h, followed by a 4-day recovery period. Left, a representative image. Scale bar: 0.5 cm. Right, biomass shown in mean ± SEM of 3 independent experiments. (G, H) Mutation in *LAO1* partially rescued the phenotypes of *atg5-1* after nitrogen starvation treatment. Four-day-old seedlings were treated with nitrogen starvation for 4 days followed by 2 days of recovery. (G) A representative image. Scale bar: 1 cm. (H) The biomass and chlorophyll content of each genotype shown in mean ± SEM of 3 independent experiments. Each independent experiment contained at least 15 seedlings. In (A), (B), (C), (F), and (H), the statistical significance was determined using one-way ANOVA (*p* < 0.05, ANOVA followed by Tukey's post hoc comparison test) analysis. Different letters denoted significant differences. Source data are available online for this figure.

starvation affects the polyubiquitination of LAO1. We transferred 7-day-old *35S: HA-LAO1* seedlings grown on 1/2 MS plates to 1/2 MS-N plates for 24 h and immunoprecipitated HA-LAO1 protein from total protein extract using anti-HA agarose beads, followed by immunoblotting with an anti-Ub antibody. We detected higher levels of polyubiquitinated HA-LAO1 after nitrogen starvation treatment (Fig. 4A), indicating that nitrogen starvation promotes the ubiquitination of LAO1.

To elucidate the mechanism underlying LAO1 autophagic degradation after nitrogen starvation, we conducted a yeast 2-hybrid (Y2H) screen using the C-terminus of LAO1 (LAO1 CT), containing the Kelch repeats, as bait. We identified that LAO1 could potentially interact with DSK2B, a substrate receptor of the selective autophagy pathway (Marshall and Vierstra, 2018; Nolan et al, 2017). For pair-wise verification, we generated two truncated forms of DSK2B, the N-terminus of DSK2B (DSK2B-NT) containing the ubiquitin-like (UBL) domain as well as the C-terminus of DSK2B (DSK2B-CT) containing the ubiquitin-associated (UBA) domain. LAO1 CT could strongly interact with DSK2B-CT but not DSK2B-NT in yeast (Fig. 4B). HA-LAO1 protein, immunoprecipitated from *35S: HA-LAO1* seedling protein extracts, could interact with recombinant His-DSK2B protein in semi-in vivo pull-down assays (Fig. 4C). Co-immunoprecipitation (co-IP) assays using *Arabidopsis* plants expressing HA-LAO1 or GFP-DSK2B and HA-LAO1 further demonstrated that LAO1 could associate with DSK2B in vivo (Fig. 4D). Furthermore, coexpression of *mCherry-LAO1* and *GFP-DSK2B* in protoplasts demonstrated clear colocalization of mCherry and GFP signals in puncta upon nitrogen starvation (Fig. 4E). These results indicate that LAO1 can physically interact with DSK2B both in vitro and in vivo.

DSK2 protein is encoded by two closely linked genes, *DSK2A* and *DSK2B*, on chromosome 2, with 87% sequence identity (Farmer et al, 2010). To determine whether DSK2 is able to regulate LAO1 abundance, we used CRISPR-Cas9 mutagenesis technology to mutate both *DSK2A* and *DSK2B* in *35S: HA-LAO1*, generating two independent HA-LAO1/*dsk2 cr* lines, #11 and #13 (Appendix Fig. S4). Immunoblot analyses revealed higher HA-LAO1 protein abundance in HA-LAO1/*dsk2 cr* #11 and #13 compared to the *35S: HA-LAO1* control under normal nitrogen

conditions (Fig. 4F,G). Additionally, nitrogen starvation-induced HA-LAO1 protein degradation was compromised in the *dsk2 cr* mutant background as well (Fig. 4F,G; Appendix Fig. S3C,D). Consistent with these observations, dual-luciferase reporter assay suggested that the relative luminescence ratio of LAO1-LUC, which is driven by either *UBQ10* promoter (UBQ), a 2 kb-long LAO1 native promoter [pLAO1(2k)], or a 3kb-long LAO1 native promoter [pLAO1(3k)], to Rennila LUC (REN), which is driven by 35S promoter, was significantly higher in *dsk2 cr* mutant protoplasts compared to Col (Fig. 4H). In contrast, overexpression of *DSK2B* (*35S: GFP-DSK2B*) significantly decreased HA-LAO1 protein level (Fig. EV3A), which could be inhibited by ConA and E64d treatment (Fig. EV3B). To investigate whether DSK2 protein level is influenced by LAO1, we raised a polyclonal antibody against DSK2. However, the immunoblot analyses indicated that the protein level of DSK2 was not changed in *35S: HA-LAO1* or *lao1*(T) mutant background compared to Col background (Fig. EV3C), indicating that LAO1 does not induce DSK2 degradation in vivo. Taken together, these results revealed that DSK2 interacts with and mediates LAO1 protein degradation during nitrogen starvation.

## Functional analysis of *DSK2* in plant fitness to nitrogen starvation

To assess whether DSK2 plays a role in plant adaptation to nitrogen starvation, we first observed the phenotypes of Col, *35S: HA-LAO1, 35S: GFP-DSK2B, 35S: GFP-DSK2B/35S: HA-LAO1* after nitrogen starvation treatment. *35S: HA-LAO1* plants show impaired growth compared to Col after nitrogen starvation treatment, while overexpression of *DSK2B* totally rescued the phenotypes of *35S: HA-LAO1* (Fig. 5A,B), consistent with DSK2-mediated LAO1 protein degradation (Fig. EV3A). We further generated two independent *dsk2 cr* lines using CRISPR-Cas9 mutagenesis technology (Fig. 5C). Immunoblot analysis indicated the absence of DSK2 protein in both *dsk2 cr* lines (Fig. 5D). We treated 4-day-old Col, HA-LAO1, *dsk2 cr* #1, *dsk2 cr* #2, HA-LAO1/*dsk2 cr* #11 and #13 seedlings with nitrogen starvation. Unexpectedly, we noticed that mutations in *DSK2* lead to better performances in both Col and HA-LAO1 backgrounds after nitrogen starvation treatment

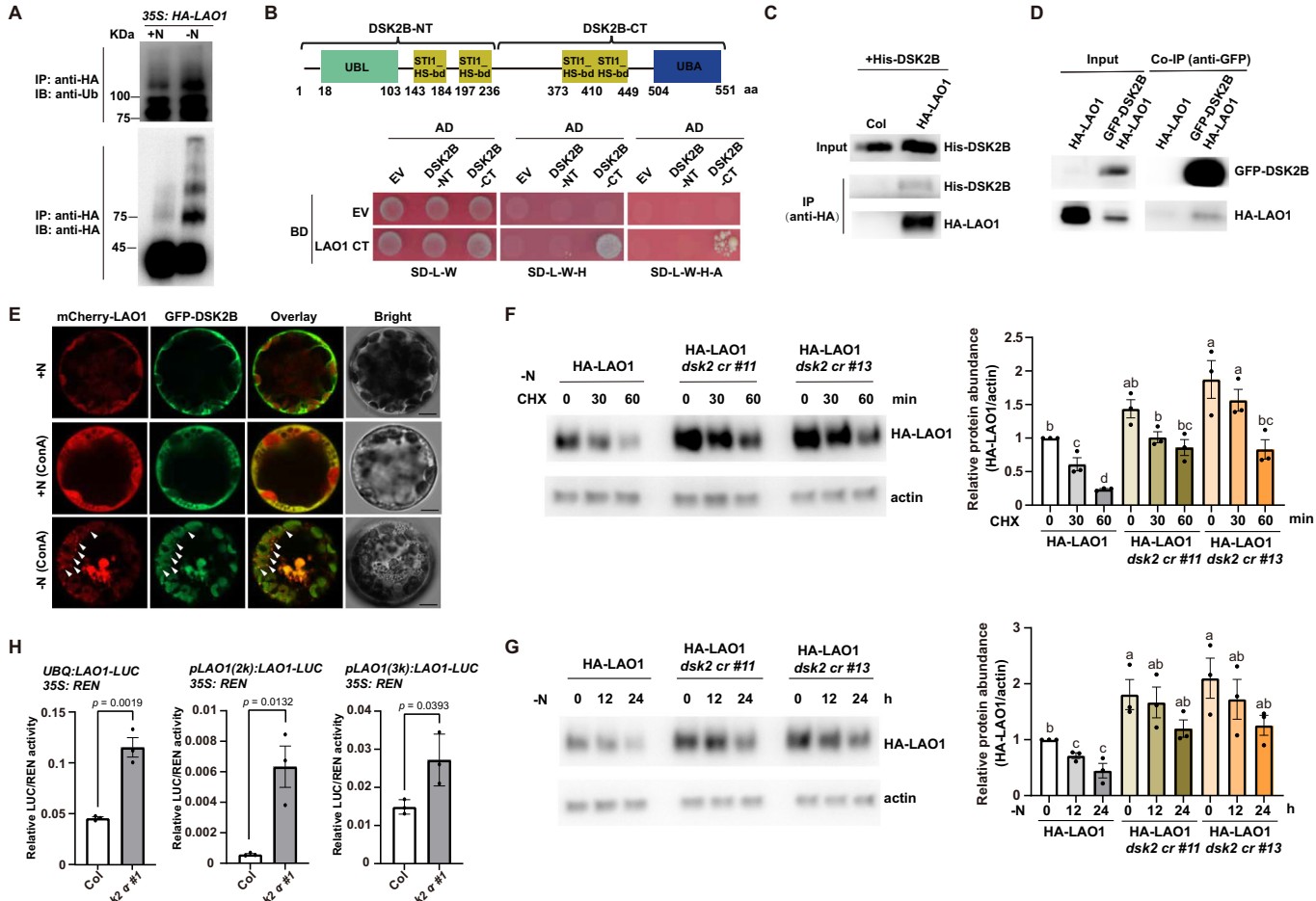

**Figure 4.    The autophagy receptor protein DSK2 interacts with and mediates nitrogen starvation-induced LAO1 protein degradation.**

(A) Detection of HA-LAO1 polyubiquitination after nitrogen-starvation treatment. Seven-day-old *35S: HA-LAO1* seedlings grown on 1/2 MS plates (+N) were treated with nitrogen starvation (-N) for 24 h before HA-LAO1 protein was immunoprecipitated using anti-HA agarose beads. The pellets were analyzed by immunoblot using anti-HA and anti-Ub antibodies. (B) LAO1 interacted with DSK2B in yeast. Corresponding constructs were co-transformed into the yeast strain Y2H-gold. The colonies appeared on SD-L-W plates were transferred onto SD-L-W-H and SD-L-W-A-H plates for interaction test. Upper showed the structure of DSK2B protein. (C) HA-LAO1 interacted with recombinant His-DSK2B in semi-in vivo pull-down assays. HA-LAO1 proteins were immunoprecipitated from transgenic *35S: HA-LAO1* plants, and then incubated with recombinant His-DSK2B protein. The pellets were boiled in 2×SDS loading buffer, and the samples were detected by western blotting using anti-His and anti-HA antibodies. (D) DSK2 co-immunoprecipitated LAO1 in *Arabidopsis*. Seven-day-old seedlings of indicated genotype were used for co-IP assays. Immunoprecipitation was performed using anti-GFP affinity beads, and bound proteins were subjected to western blotting using anti-GFP and anti-HA antibodies. (E) mCherry-LAO1 colocalized with GFP-DSK2B in puncta following nitrogen starvation treatment. Protoplasts were isolated and transfected with constructs expressing *35S: mCherry-LAO1* and *35S: GFP-DSK2B*. +N, incubation medium containing 5 mM nitrate. +N (ConA), incubation medium containing 5 mM nitrate and 1 μM ConA. -N (ConA), nitrogen-free incubation medium containing 1 μM ConA. Scale bars: 5 μm. (F, G) HA-LAO1 proteins were more stable in the *dsk2* mutant background. Seven-day-old *35S: HA-LAO1, 35S: HA-LAO1/dsk2 cr #11*, and #13 seedlings grown on 1/2 MS plates (+N) were subjected to nitrogen starvation, with (F) or without (G) CHX, for the indicated times, and total proteins were extracted. Western blots were performed using anti-HA and anti-actin antibodies. Left, representative western blot results. Right, relative HA-LAO1 protein levels shown in mean ± SEM of 3 independent experiments. The statistical significance was determined using one-way ANOVA ($p < 0.05$, ANOVA followed by Tukey's post hoc comparison test) analysis. Different letters denoted significant differences. (H) The luminescence ratio of LAO1-LUC/REN in Col and *dsk2 cr* mutant protoplasts. The experiment design was similar as Fig. 3D, except that three different promoters were used to drive LAO1-LUC coding sequence. Relative LUC/REN activity shown in mean ± SEM of 3 independent experiments. All the statistical significances were calculated by the Student's *t* test. Source data are available online for this figure.

(Fig. 5E,F). RT-qPCR analyses indicated that the transcript levels of *LAO1* in *dsk2 cr* #1 were comparable to those in Col before nitrogen starvation treatment, but slightly higher after nitrogen starvation (Fig. EV3D), ruling out the possibility that the improved performance of *dsk2 cr* mutant was attributed to low expression levels of *LAO1*. These results not only indicate that DSK2-mediated LAO1 protein degradation participates in plant tolerance to nitrogen starvation, but also suggest that DSK2 may also control

the protein level of a factor that antagonizes the effect of LAO1 protein accumulation.

## DSK2 interacts with and degrades a group of class I TCP transcription factors

To identify this potential factor, we conducted a yeast 2-hybrid screening using DSK2B as bait. We found that DSK2B could

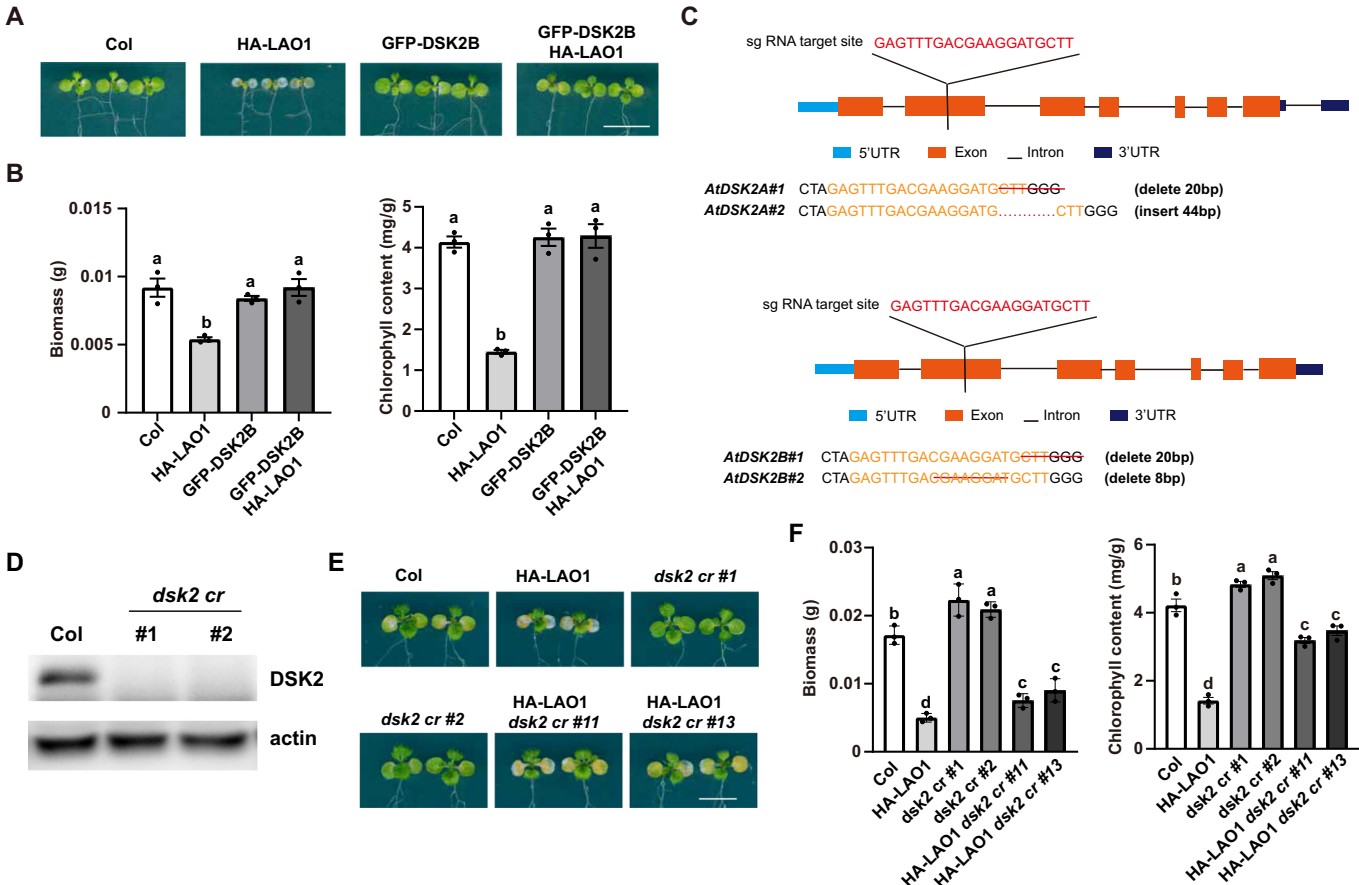

**Figure 5. DSK2 participates in LAO1-mediated plant responses to nitrogen starvation.**

(A, B) Overexpression of *DSK2B* rescued the phenotype of *35S: HA-LAO1* (*LAO1* OE1) plants after nitrogen starvation treatment. Four-day-old seedlings of indicated genotype were transferred to nitrogen starvation condition for 4 days, and then transferred back to normal nitrogen condition for another 2 days to recover. (A) Representative images. Scale bar: 1 cm. (B) The biomass and chlorophyll content of each genotype shown in mean ± SEM of 3 independent experiments. Each independent experiment contained at least 15 seedlings. (C) Mutation types induced by CRISPR-Cas9 at *DSK2A* and *DSK2B* gene loci. The gene structure of *DSK2A* and *DSK2B* and the position of sgRNA were shown. Mutation was confirmed by PCR-sequencing of targeted locus. (D) Confirmation of the two *dsk cr* mutant lines using endogenous DSK2 antibody. Total proteins were extracted from 7-day-old Col and the two *dsk cr* mutant seedlings. Western blots were performed using anti-DSK2 and anti-actin antibodies. (E) Seedling phenotypes of indicated genotype after nitrogen starvation for 6 days followed by 4 days of recovery. Scale bar: 1 cm. (F) The biomass and chlorophyll content of each genotype shown in mean ± SEM of 3 independent experiments. Each independent experiment contained at least 15 seedlings. All statistical significances were determined using one-way ANOVA analysis ($p < 0.05$, ANOVA followed by Tukey's post hoc comparison test). Different letters denoted significant differences. Source data are available online for this figure.

potentially interact with a group of class I TCP transcription factors, TCP7, TCP14, TCP15, and TCP21. We verified the interaction in yeast via pair-wise verifications (Fig. 6A). In vitro pull-down assays indicate that GST-DSK2B CT, but not GST alone, could pull down recombinant His-TCP7, His-TCP14, His-TCP15, and His-TCP21 proteins (Fig. 6B). Co-IP assays confirmed the interaction of both GFP-TCP15 and GFP-TCP21 with DSK2 in vivo (Fig. 6C). Altogether, our data suggest that DSK2 can interact with a group of class I TCPs both in vitro and in vivo.

Subsequently, we investigated whether DSK2 could also facilitate TCP protein degradation. Protoplasts were isolated from *35S: GFP-TCP21* transgenic plants, and constructs for DSK2B mRNA interference (*DSK2B*-RNAi) or *DSK2B* overexpression (*DSK2B*-OE) along with their respective empty vectors were transiently expressed in these protoplasts. Immunoblot analyses indicated that *DSK2B*-RNAi increased GFP-TCP21 protein level, whereas *DSK2B*-

OE decreased it (Fig. 6D). Furthermore, protoplasts were isolated from Col and *dsk2 cr* mutant plant leaves, and the construct expressing TCP7-LUC, TCP14-LUC, TCP15-LUC, or TCP21-LUC, driven by UBQ promoter, was, respectively, transfected in these protoplasts. Dual-luciferase reporter assay suggested that the luminescence ratios of TCPs-LUC/REN in the protoplasts of *dsk2 cr* mutant were significantly higher than in those of Col (Fig. 6E). Both MG132 and E64d treatments interfered with DSK2B-mediated TCP14 and TCP21 protein degradation (Fig. 6F), indicating that both ubiquitin-proteasome system and autophagy play roles in DSK2-mediated TCP degradation. Furthermore, TCP14 and TCP21 proteins accumulated after nitrogen starvation, which was more prominent in the *dsk2 cr* mutant background (Fig. 6G). Taken together, the results above indicate that DSK2 mediates the degradation of a group of class I TCP transcription factors by directly interacting with them.

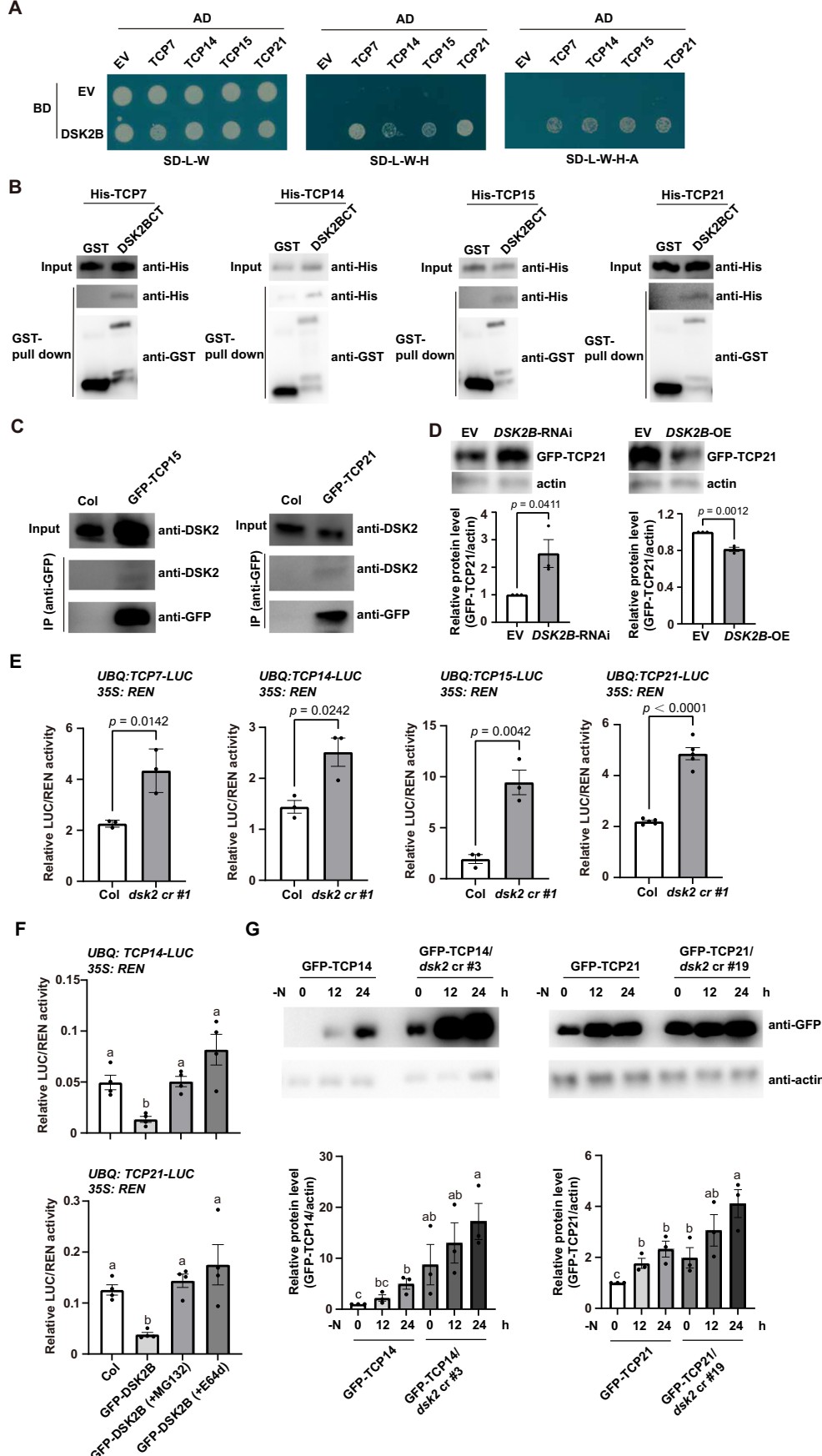

**Figure 6. DSK2 interacts with and degrades a group of class I TCP transcription factors.**

(A) DSK2B interacted with TCPs in yeast. Corresponding constructs were co-transformed into the yeast strain Y2H-gold. SD-L-W-H and SD-L-W-A-H plates were used for interaction test. (B) In vitro GST pull-down assays. GST or GST-DSK2BCT were incubated with His-TCP7, His-TCP14, His-TCP15, or His-TCP21, respectively. The pellets were analyzed by immunoblot using anti-GST and anti-His antibodies. (C) TCP15 and TCP21 co-immunoprecipitated DSK2 in *Arabidopsis*. Seven-day-old seedlings of indicated genotype were used for co-IP assays. Immunoprecipitation was performed using GFP_Trap beads, and bound proteins were subjected to western blotting using anti-GFP and anti-DSK2 antibodies. (D) Immunoblot analysis of the protein abundance of GFP-TCP21. Protoplasts were isolated from *35S: GFP-TCP21* plants. Constructs for *DSK2B*-RNAi or *DSK2B*-OE were transfected, respectively. Total proteins were extracted from transfected protoplasts, and analyzed by western blots using anti-HA and anti-actin antibodies. Upper, representative western blot results. Lower, relative GFP-TCP21 protein levels shown in mean ± SEM of 3 independent experiments. Statistical significances were calculated by the Student's *t* test. (E) The luminescence ratio of TCP-LUC/REN in Col and *dsk2 cr* mutant protoplasts. Relative TCP-LUC/REN activity was shown in mean ± SEM of 3 independent experiments. Statistical significances were calculated by the Student's *t* test. (F) Both MG132 and E64d inhibited DSK2-mediated turnover of TCP proteins. Protoplasts were isolated from Col and *35S:GFP-DSK2B* plants and transfected with the indicated constructs. Transfected protoplasts were treated with 50 μM MG132 or 50 μM E64d for 4 h before relative TCP-LUC/REN was determined. DMSO was used as negative control. Data were shown in mean ± SEM of 3 independent experiments. (G) Protein levels of GFP-TCP14 and GFP-TCP21 during nitrogen starvation treatment. Seven-day-old seedlings of the indicated genotypes were subjected to nitrogen starvation for the indicated times. Total proteins were extracted and analyzed by immunoblot using anti-GFP and anti-actin antibodies. Upper, a representative immunoblot result; lower, relative protein levels shown in mean ± SEM of 3 independent experiments. In (F) and (G), statistical significances were determined using one-way ANOVA analysis (*p* < 0.05, ANOVA followed by Tukey's post hoc comparison test). Different letters denoted significant differences. Source data are available online for this figure.

## DSK2-TCP module participates in LAO1-mediated nitrogen starvation response

As DSK2 can interact with and mediate the degradation of both LAO1 and class I TCPs, we further asked whether class I TCPs are able to antagonistically regulate plant response to nitrogen starvation relative to LAO1. Multiple *tcp* single, double, triple, and higher order mutants were obtained or generated (Appendix Fig. S5A), and we observed their performances after nitrogen starvation treatment. While *tcp14*, *tcp15*, and *tcp14/15* mutant seedlings showed no significant differences compared to Col after nitrogen starvation treatment, *tcp7/14/21*, *tcp14/15/21*, *tcp7/14/15/21* (*tcpQ*), and *tcp septuple* (*tcpS*) mutant seedlings showed significantly reduced biomass and chlorophyll contents compared to Col (Fig. 7A,B). To provide further evidence, we generated overexpression lines of *TCP14* (*TCP14* OE #32) and *TCP21* (*TCP21* OE #77; Appendix Fig. S6A). Overexpression of *TCP14* and *TCP21* improved plant fitness to nitrogen starvation (Appendix Fig. S6B,C). We then crossed these lines with *dsk2 cr* #1, resulting in the generation of *TCP14* OE/*dsk2 cr* #3 and 35S: *TCP21* OE/*dsk2 cr* #19, respectively. However, overexpression of *TCP14* and *TCP21* failed to further enhance the *dsk2 cr* mutant phenotypes (Appendix Fig. S6B,C), suggesting that overexpression of a single *TCP* gene might be insufficient to significantly enhance the effect beyond what is already achieved by the coordinated action of multiple TCPs accumulating in the *dsk2 cr* mutant. We further generated *dsk2 cr tcpQ* mutant by introducing mutations in *DSK2A* and *DSK2B* via CRISPR-Cas9 technology in *tcpQ* mutant background (Appendix Fig. S7A). The performances of the *dsk2 cr tcpQ* after nitrogen starvation treatment resembled that of the *tcpQ* mutant (Appendix Fig. S7B,C), indicating that *TCPs* are epistatic to *DSK2*. These results support that class I TCPs are positive regulators of plant tolerance to nitrogen starvation, which functions downstream of DSK2. However, the expression levels of *DSK2B* and *LAO1* in *tcp* mutants or *TCP* overexpression lines were comparable to those in Col (Fig. EV4A), suggesting that the nitrogen starvation intolerant phenotypes of *tcp* mutants are not attributed to the misregulation of *DSK2* or *LAO1* transcript levels.

Furthermore, we analyzed the regulatory and genetic relationships between class I TCPs and LAO1 with respect to plant tolerance to nitrogen starvation. Although LAO1 could interact

with TCPs, there was no evidence to support that LAO1 could target TCPs for degradation (Fig. EV4B–F). Nonetheless, we found that overexpression of either *TCP14* or *TCP21* in *35S: HA-LAO1* background could almost totally rescue the nitrogen starvation sensitivity of *35S: HA-LAO1* (Fig. 7C,D), suggesting that TCPs may be epistatic to LAO1. To this end, we generated *lao1 cr tcpQ* mutant by introducing mutation in *LAO1* gene via CRISPR-Cas9 technology in *tcpQ* mutant background (Appendix Fig. S5B). The performances of *lao1 cr tcpQ* after nitrogen starvation treatment resembled that of *tcpQ* mutant (Fig. 7E,F), indicating that class I TCPs function downstream of LAO1 genetically to facilitate plant performance after nitrogen starvation. Taken together, these observations suggest that class I TCPs are positive regulators of plant adaptation to nitrogen starvation, functioning genetically downstream of *LAO1*.

## Discussion

Proper responses to abiotic stresses are crucial for plant development and survival. Our results indicate a complex regulatory network comprising protein stability regulators and transcription factors that govern plant tolerance to nitrogen starvation. In this model, the F-box protein LAO1 primarily compromises plant performance following nitrogen starvation, while class I TCPs mitigate the effect of LAO1. DSK2, a master ubiquitin receptor protein, orchestrates the degradation of both LAO1 and class I TCPs to fine-tune plant tolerance to nitrogen starvation (Fig. EV5).

### The F-box protein LAO1 is a novel regulator of plant adaptation to nitrogen starvation

Nitrogen is an important macronutrient for plant development, yet plants, including major crops, frequently encounter nitrogen deficiency during their life cycles. Adequate and precise responses to nitrogen starvation stress are essential for plant survival and crop yield. The ubiquitin-proteasome system has been shown to regulate plant nitrogen homeostasis (Mackinnon and Stone, 2022), but the involvement and role of F-box proteins in this regulation remain unclear. In this study, we identified a previously uncharacterized F-box protein LAO1 as a novel negative regulator of plant nitrogen

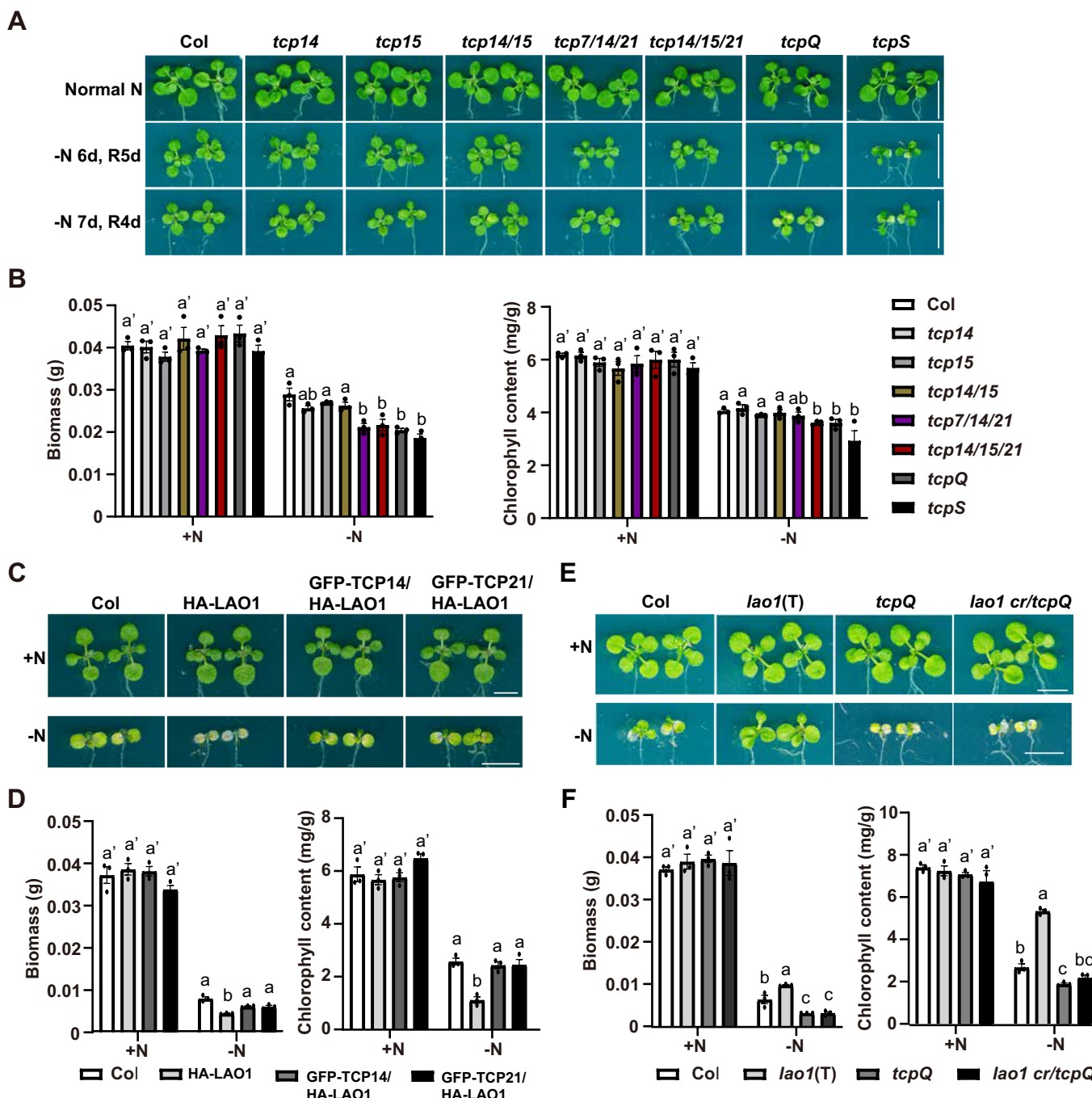

**Figure 7. Roles of TCP transcription factors in plant response to nitrogen starvation.**

(A, B) Phenotypes of multiple *tcp* mutants after nitrogen starvation. Seedlings of the indicated genotype were grown on 1/2 MS plates for 4 days, and then were transferred to 1/2 MS-N plates either for another six days followed by five days of recovery on 1/2 MS plates (-N 6 d, R 5 d) or for another 7 days followed by 4 days of recovery on 1/2 MS plates (-N 7 d, R 4 d). (A) Representative images. Scale bars: 1 cm. (B) The biomass and chlorophyll content of each genotype under the -N 7 d, R 4 d treatment in (A) were presented in mean ± SEM of 3 independent experiments. Each independent experiment contained at least 15 seedlings. (C, D) Overexpression of *TCP14* or *TCP21* rescued the phenotype of *35S: HA-LAO1* after nitrogen starvation treatment. Four-day-old seedlings of indicated genotype were transferred to nitrogen starvation condition for 4 days, and then transferred back to normal nitrogen condition for another 2 days to recover. (C) Representative images. Scale bars: 0.5 cm. (D) The biomass of each genotype shown in mean ± SEM of 3 independent experiments. Each independent experiment contained at least 15 seedlings. (E, F) Genetic analysis of *LAO1* and *TCP* during nitrogen starvation. Four-day-old seedlings were treated with nitrogen starvation for 6 days followed by 4 days of recovery. (E) Representative images. Scale bars: 0.5 cm. (F) The biomass and chlorophyll content of each genotype shown in mean ± SEM of 3 independent experiments. Each independent experiment contained at least 15 seedlings. All statistical significance was determined using one-way ANOVA ($p < 0.05$, ANOVA followed by Tukey's post hoc comparison test) analysis. Different letters denoted significant differences. Source data are available online for this figure.

starvation adaptation (Fig. 1). What's more, we also observed that LAO1 abundance is associated with developmental processes involving nitrogen assimilation or remobilization (Fig. EV2), indicating a broader role of LAO1 in plant response to environmental and in vivo nitrogen level dynamics. We further provide evidences that plants deal with nitrogen starvation by inhibiting *LAO1* transcription and facilitating autophagic turnover of LAO1 protein under nitrogen-deficient conditions (Figs. 2 and 3). These results suggest a complex regulation mechanism between N availability and LAO1 protein abundance, involving both developmental stage and the abiotic stress of N starvation. While our results highlighted the significance of transcriptional repression and autophagic degradation of LAO1 during nitrogen starvation, the findings in Fig. EV2 suggest that LAO1 protein levels may require a coordinated balance between synthesis and degradation to adapt to fluctuating nitrogen levels under natural environment. Nitrogen starvation decreased GUS signals in both *proLAO1(3k):-GUS* and *proLAO1(2k):GUS* seedlings; however, the average GUS signals in *proLAO1(3k):GUS* lines were stronger than those in *proLAO1(2k):GUS* (Fig. 2B). This result indicates that, while both *proLAO1(3k)* and *proLAO1(2k)* are responsive to nitrogen starvation, additional key cis-elements may exist in *proLAO1(3k)* that are crucial for *LAO1* expression in vivo.

## Ubiquitin-mediated protein turnover pathways coordinate plant response to nitrogen starvation

Ubiquitinated proteins are mainly degraded through ubiquitin-proteasome and autophagy pathways. Several recent studies have shed light on the significance of the interplay between autophagy and the 26S proteasome in both stress tolerance and silique developmental processes (Marshall et al, 2015; Yu and Hua, 2022). Our results reveal an additional level of interaction between these two major pathways that control protein homeostasis in plant cells during plant adaptation to nitrogen starvation, indicating a more intricate coordination of autophagy and the UPS pathways during plant-environment interaction.

Our results suggest that LAO1 is able to physically interact with DSK2. The interaction between LAO1 and DSK2 presents two potential scenarios. Firstly, LAO1 may ubiquitinate and degrade DSK2, thereby disrupting autophagy function and compromising *Arabidopsis* fitness under nitrogen starvation. Secondly, DSK2 could mediate the degradation of LAO1 protein via selective autophagy after nitrogen starvation, helping plants to survive nitrogen starvation. However, immunoblot analyses demonstrate that DSK2 protein levels remain unchanged in *35S: HA-LAO1* background relative to Col background (Fig. EV3C), indicating that LAO1 does not induce DSK2 protein degradation in vivo. Conversely, our data support that LAO1 protein is regulated by DSK2-mediated autophagic degradation under nitrogen starvation conditions (Figs. 4 and EV3A,B). The autophagic degradation of LAO1 is also supported by the observation that mutation in *LAO1* could improve the performance of *atg5-1* mutant after nitrogen starvation (Fig. 3G,H). It is important to note, however, that *LAO1* mutation did not fully rescue the phenotype of *atg5-1* mutant, indicating that LAO1 is not the exclusive target of autophagy under nitrogen starvation. Considering that autophagy pathway recycles nutrients by degrading dispensable components in cells both in bulk and selectively (Marshall and Vierstra, 2018), there may be additional factors, downstream of the autophagy pathway, that are

critical for plant adaptation to nitrogen starvation. What's more, neither the autophagy inhibitor E64d nor *atg* mutations completely abolished nitrogen starvation-induced LAO1 protein degradation (Fig. 3A–C), suggesting that LAO1 may be destabilized by nitrogen starvation through alternative pathways.

## Class I TCP transcription factors are novel positive regulator of plant responses to nitrogen starvation

Our data support a critical role of class I TCP transcription factors in plant adaptation to nitrogen starvation. As Class I TCPs act genetically downstream of LAO1 to counteract its effects (Fig. 7C–F), it is plausible that the F-box protein LAO1 could degrade these TCPs. However, our attempts to clarify this possibility reveal that, while LAO1 can interact with TCPs, it may not mediate the degradation of TCP proteins (Fig. EV4). These findings suggest that LAO1 and TCPs may regulate distinct but related aspects of plant nitrogen starvation responses or LAO1 might interfere with the activity of TCP proteins. We found that DSK2 could interact with and target class I TCPs for degradation as well (Fig. 6). The regulatory module of DSK2-TCP and the genetic hierarchy between LAO1 and TCP explain why *DSK2* knockout mutations did not manifest *LAO1* overexpression phenotypes (Fig. 5). Genetic analyses further support that TCPs function downstream of DSK2 in the regulation of N starvation tolerance (Appendix Fig. S7). In addition, the positive role of TCPs in nitrogen starvation tolerance aligns with prior findings that TCP20 interacts with NLP6&7 to activate the expression of nitrate assimilation and signaling genes (Guan et al, 2017). Moreover, OsTCP19, a close homolog to *Arabidopsis* TCP14 and TCP15, regulates tillering in response to nitrogen levels and controls nitrogen use efficiency in rice (Liu et al, 2021). These findings not only suggest that class I TCPs can participate in plant response to nitrogen starvation stress, but also reinforce the potential of genetic manipulation of class I TCP transcription factors to improve crop performance under fluctuating soil nitrogen conditions.

# Methods

**Reagents and tools table**

| Reagent/Resource | Reference or Source | Identifier or Catalog Number |
|---|---|---|
| **Experimental models** | | |
| *Arabidopsis*: Col-0 | N/A | N/A |
| *Arabidopsis*: 35S::HA-LAO1 | This study | N/A |
| *Arabidopsis*: 35S::GFP-LAO1 | This study | N/A |
| *Arabidopsis*: 35S::GFP-DSK2 | This study | N/A |
| *Arabidopsis*: 35S:: HA-LAO1/ *atg5-1* | This study | N/A |
| *Arabidopsis*: 35S:: HA-LAO1/ *atg7-3* | This study | N/A |
| *Arabidopsis*: *tcp14* | Wang et al, 2023 | N/A |

| Reagent/Resource | Reference or Source | Identifier or Catalog Number |
|---|---|---|
| *Arabidopsis*: tcp15 | Wang et al, 2023 | N/A |
| *Arabidopsis*: tcp14/15 | Peng et al, 2015 | N/A |
| *Arabidopsis*: tcp7/14/21 | This study | N/A |
| *Arabidopsis*: tcp14/15/21 | This study | N/A |
| *Arabidopsis*: tcpQ | This study | N/A |
| *Arabidopsis*: tcpS | Zhang et al, 2018 | N/A |
| *Arabidopsis*: 35S::GFP-TCP14/HA-LAO1 | This study | N/A |
| *Arabidopsis*: 35S::GFP-TCP21/HA-LAO1 | This study | N/A |
| *Arabidopsis*: 35S::GFP-DSK2/HA-LAO1 | This study | N/A |
| *Arabidopsis*: 35S:: HA-LAO1/ dsk2 cr | This study | N/A |
| *Arabidopsis*: lao1 (T)/ atg5-1 | This study | N/A |
| **Recombinant DNA** | | |
| pGEX-4T-1-DSK2CT | This study | N/A |
| pET32a-TCP7 | This study | N/A |
| pET32a-TCP14 | This study | N/A |
| pET32a-TCP15 | This study | N/A |
| pET32a-TCP21 | This study | N/A |
| pET32a-DSK2 | This study | N/A |
| **Antibodies** | | |
| Mouse anti-GST | Affinity | Cat# T0007 |
| Rabbit anti-DSK2 | Homemade | N/A |
| Mouse anti-His | Abmart | Cat# M30111 |
| Mouse anti-GFP | Abmart | Cat# M20004 |
| Rabbit anti-plant actin | Easybio | Cat# BE0027 |
| Anti-Mouse IgG Peroxidase antibody | Sigma | Cat# A9044 |
| Anti-Rabbit IgG Peroxidase antibody | Sigma | Cat# A9169 |
| Anti-HA | Roche | Cat# 12013819001 |
| Rabbit anti-UB | ProteinTech | Cat# 10201-2-AP |
| **Oligonucleotides and other sequence-based reagents** | | |
| PCR Primers | This study | Appendix Table S1 |
| **Chemicals, Enzymes and other reagents** | | |
| Murashige and Skoog Basal Medium | PhytoTech | Cat# M519 |
| MS Base Salts (-N, with vitamins) | Coolaber | Cat# PM1011 |
| Hygromycin B | Mei5bio | Cat# H61237 |
| DMSO | BBI | Cat# 67-68-5 |
| protease inhibitor cocktail | MCE | Cat# HY-K0010 |

| Reagent/Resource | Reference or Source | Identifier or Catalog Number |
|---|---|---|
| Magnesium chloride hexahydrate | aladdin | Cat# 7791-18-6 |
| Glycerol | BBI | Cat# A600232-0500 |
| Triton X-100 | Yeasen | Cat# 20107ES76 |
| NP40 | Yeasen | Cat# 20103ES60 |
| PMSF | Diamond | Cat# A100754-005 |
| Urea | BBI | Cat# A510907-002 |
| $NaH_2PO_4$ | Fisher Scientific | Cat# S369 |
| 2-mercaptoethanol | MACKLIN | Cat# CC16419978 |
| NON-Fat Powdered milk | BBI | Cat# A600669-0250 |
| MG132 | MCE | Cat# S21619 |
| TBST | Fdbio | Cat# FD9061 |
| Tris | BBI | Cat# 77-86-1 |
| SDS | BBI | Cat# 151-21-3 |
| Glycine | BBI | Cat# 56-40-5 |
| Cycloheximide | beibokit | Cat# A1740A |
| TEMED | Fdbio | Cat# FD2100 |
| IPTG | BBI | Cat# 367-93-1 |
| **Software** | | |
| GraphPad Prism 9 | GraphPad Software, USA | https://www.graphpad.com/scientific-software/prism/ |
| ImageJ | NIH, USA | https://imagej.nih.gov/ij |
| Adobe Illustrator 2023 | Adobe | N/A |
| **Other** | | |
| Anti-HA Affinity Beads | Smart-Lifesciences | Cat# SA068001 |
| Anti-GFP Affinity Beads | Smart-Lifesciences | Cat# SA070001 |
| Ni-NTA Sepharose 6FF | BBI | Cat# C600033-0010 |
| Glutathione Agarose Resin | Yeasen | Cat# 20507ES10 |
| FDbio-Dura ECL | FdBio | Cat# FD8020 |
| ClonExpress II One Step Cloning Kit | Vazyme | Cat# C112-01 |

## Plant materials and growth conditions

All the *Arabidopsis* genetic materials used in this study were Columbia (Col-0) ecotype. The *lao1*(T) mutant seeds (CS900954) were obtained from ABRC; the *atg5-1* and *atg7-3* (SAIL_129_B07) mutant seeds were gifted by Dr. Ronghui Pan. The *tcp14-2 (tcp14)*, *tcp15-1 (tcp15)* was described in Wang et al, 2023. The *tcp14-3 tcp15-3 (tcp14/15)* and *tcpS* mutant seeds were described in Peng et al, 2015 and Zhang et al, 2018, respectively. The *dsk2 cr*, *tcpQ*, *nia2 cr*, *nia2 cr/HA-LAO1*, and *lao1 cr/tcpQ* mutant plants were obtained by CRISPR-Cas9 induced mutagenesis, and the mutations were confirmed by PCR-based sequencing. The *tcp7/14/21* and

*tcp14/15/21* were isolated from the genetic cross of *tcpQ* with Col, and the mutations were confirmed by PCR-based sequencing. The *35S: GFP-DSK2B, 35S: GFP-LAO1, 35S: HA-LAO1, 35S: GFP-TCP14, 35S: GFP-TCP21* transgenic and the *proLAO1(3k): GUS,* and *proLAO1(2k): GUS* transgenic plants were obtained by transforming agrobacteria GV3101 harboring respective construct into Col-0 background. The *35S: GFP-DSK2B /35S: HA-LAO1, 35S: GFP-TCP14/35S: HA-LAO1,* and *35S: GFP-TCP21 /35S: HA-LAO1* were obtained by transforming agrobacteria GV3101 containing *35S: GFP-DSK2B, 35S: GFP-TCP14* and *35S: GFP-TCP21,* respectively, into *35S: HA-LAO1* background. Transgenic plants were selected based on the relevant antibiotic resistance and RT-qPCR analysis. The *atg5-1 lao1*(T) double mutant was obtained by genetic cross of *atg5-1* and *lao1*(T). The *35S: HA-LAO1/atg5-1* and *35S: HA-LAO1/atg7-3* genetic materials were obtained by genetic cross of *35S: HA-LAO1* with *atg5-1* and *atg7-3,* respectively. The *GFP-TCP21/lao1(T)* and *GFP-TCP21* genetic materials were obtained through genetic cross of *GFP-TCP21/HA-LAO1* with *lao1(T).*

Adult *Arabidopsis* plants were grown in a growth chamber under a long-day condition (16 h light/8 h dark) at 22 °C. For experiments using seedlings, seeds were sterilized with 8% NaClO, stratified at 4 °C in the dark for 2 days, and then germinated onto half-strength Murashige and Skoog ($^1/_2$ MS) plates containing both nitrate and ammonium as nitrogen resources, pH 5.7. The seedlings were grown under continuous white light condition at 22 °C in a growth chamber for indicated time followed by different downstream treatments. For nitrogen starvation phenotypes, 4-day-old seedlings of each genotype were transferred onto $^1/_2$ MS-N agar plates lacking both nitrate and ammonium for indicated time following recovery. For RT-qPCR analyses, 4-day-old Col seedlings were transferred to $^1/_2$ MS-N agar plates for 12, 24, 36, 48, 72, and 96 h, respectively, for total RNA extraction. For western blots, 7-day-old indicated genotype seedlings were transferred to $^1/_2$ MS-N agar plates for indicated time, for total protein extraction.

## Vector constructions

For *35S: HA-LAO1* transgenic lines, the coding sequence (CDS) of *LAO1* fused with N-terminal flag tag was cloned into pJim19(bar) 3×HA backbone driven by 35S promoter.

For *35S: GFP-LAO1, 35S: GFP-DSK2B, 35S: GFP-TCP14* and *35S: GFP-TCP21* overexpressing lines, the full-length coding sequences (CDSs) were amplified from *Arabidopsis* cDNA and cloned into pCAMBIA1300 expression vector using CloneExpress (Vazyme). For *lao1 cr/tcpQ, nia2cr, nia2 cr/HA-LAO1, dsk2 cr,* and *tcpQ* mutants, the sgRNAs were designed using CRISPR-P website (http://crispr.hzau.edu.cn/cgi-bin/CRISPR/CRISPR), and the final expression vector was generated following a typical cloning process that was described previously (Wang and Chen 2020).

For LAO1 and DSK2 Y2H experiments, the CDS of *LAO1 CT* (86 to 377 a.a.) was inserted into the pGBKT7 (BD) vector to generate BD-LAO1 CT. The CDSs of *DSK2B*-NT (1 to 236 a.a.) and *DSK2B*-CT (237 to 552 a.a.) were inserted into the pGADT7(AD) vector to generate AD-DSK2B-NT and AD-DSK2B-CT, respectively. For DSK2 and TCPs Y2H experiments, the CDS of *DSK2B* was inserted into the pGBKT7 (BD) vector to generate BD-DSK2B. The CDSs of *TCP7, TCP14, TCP15,* and *TCP21* were inserted into the pGADT7(AD) vector to generate AD-TCP7, AD-TCP14, AD-TCP15, and AD-TCP21, respectively. Corresponding constructs

were transformed into the yeast strain Y2H gold for Y2H assay. For LAO1 and TCPs Y2H experiments, the CDS of *LAO1 CT* was inserted into the pGBKT7 (BD) vector to generate BD-LAO1CT. The CDSs of *TCP7, TCP14, TCP15,* and *TCP21* were inserted into the pGADT7(AD) vector to generate AD-TCP7, AD-TCP14, AD-TCP15, and AD-TCP21, respectively. Corresponding constructs were transformed into the yeast strain Y2H gold for Y2H assay.

For recombinant protein expression and purification, the CDSs encoding *DSK2B, TCP7, TCP14, TCP15,* and *TCP21* were cloned into BamHI/SalI-digested pET32a to generate His-DSK2B, His-TCP7, His-TCP14, His-TCP15, and His-TCP21 construct, while the CDS encoding *DSK2BCT* was cloned into EcoRI-digested pGEX-4T-1 to generate GST-DSK2BCT, and the resulting constructs were transformed into Rosetta (DE3) strain for downstream applications.

For the dual-LUC reporter assay, the *UBQ10* promoter was cloned into pGreen II-0800-Luc vector, yielding pGreenII-0800-UBQ-LUC. Then the coding region of *LAO1, TCP7, TCP14, TCP15,* and *TCP21* were cloned into the pGreenII-0800-UBQ-LUC vector. For LAO1-LUC driven by *LAO1* native promoter, the *UBQ10* promoter in the vector was replaced with *LAO1* native promoter. All the resulting constructs were transformed into protoplast for downstream applications.

For *DSK2* RNAi, two *DSK2B* coding fragments (nucleotides 384 to 1024 and 1102 to 1655 as measured from the first base of the start codon) were amplified by PCR using primer pairs dsDSK2B-1F/dsDSK2B-1R and dsDSK2B-2F/dsDSK2B-2R. Each of these fragments was subsequently cloned in the reverse orientation and separated by the intron sequence of the *Petunia hybrida* Chalcone Synthase gene in pFGC5941, yielding *DSK2B*-RNAi construct.

For mCherry-LAO1, mCherry fragment was cloned into the pCAMBIA 35S: GFP vector to replace GFP coding sequence, yielding pCAMBIA 35S: mCherry vector. Subsequently, the CDS encoding *LAO1* was cloned into the pCAMBIA 35S: mCherry vector to obtain mCherry-LAO1.

For bimolecular fluorescence complementation assay, the full-length coding sequences of *LAO1* and that of the indicated *TCPs* (*TCP14* and *TCP21*) were cloned in-frame of the sequence encoding the N-terminal half of YFP (YN) to obtain *LAO1-YN,* and with the sequence encoding the C-terminal half of YFP (YC) to obtain *TCP14-YC* and *TCP21-YC,* respectively.

The primer sequences used for molecular cloning were listed in Appendix Table S1.

## Western blot

Protein samples were first resolved via SDS-PAGE, and then transferred onto a PVDF film using Trans-Blot Turbo (Bio-Rad). After being blocked with 5% milk in TBS-T, the film was incubated with primary antibodies at 4 °C overnight and then probed with corresponding secondary antibody. The signal was captured by MiniChemi (Beijing Sage Creation Science Co, LTD). The primary antibodies used in this study were anti-actin (1:4000, EASYBIO), anti-GST (1:1000, Affinity), anti-His (1:1000, Abmart), anti-GFP (1:1000, Abmart), anti-HA (1:1000, Abmart), anti-DSK2 (1:500, home-made). The secondary antibodies used were Goat Anti-Mouse IgG (whole molecule)-Peroxidase antibody (1:10,000, Sigma) or Goat Anti-Rabbit IgG (whole molecule)-Peroxidase antibody (1:10,000, Sigma).

## Gene expression analysis

Total RNA was extracted using Plant RNA kit (Omega). 500 ng total RNA was used for first-strand cDNA synthesis using M5 Super plus qPCR RT kit with gDNA remover (Mei5bio). The cDNA was diluted 10 folds, and 1 μL was used as template for quantitative polymerase chain reaction (qPCR) using 2×M5 HiPer SYBR Premix EsTaq (Mei5bio). *ACT7* was used as internal control. The primer sequences used for qPCR were listed in Appendix Table S1.

## Yeast two-hybrid (Y2H) assays

The Y2H screen was conducted using C-terminal of *LAO1* (pGBKT7-LAO1 CT) or *DSK2B* (pGBKT7-DSK2B) as bait. The pGBKT7-LAO1 CT or pGBKT7-DSK2B construct was co-transformed with the *Arabidopsis* cDNA library in pGADT7 vector into Y2H-Gold yeast strain. Positive colonies were picked out for PCR-based sequencing to identify potential interacting partners. The interaction was first verified via pair-wise Y2H experiment. Yeast colonies appeared on Double DO supplement (-Leu/-Trp, SD-L-W) plates were further transferred onto Triple DO supplement (-Leu/-Trp/-His, SD-L-W-H) or Quadruple DO supplement (-Leu/-Trp/-Ade/-His, SD-L-W-H-A) for interaction test.

## Extraction of plant total protein

Typically, similar amounts of seedlings were ground into fine powder in liquid nitrogen using a homogenizer, and then 100 μL denaturing buffer (8 M urea, 100 mM $NaH_2PO_4$, 100 mM Tris-HCl, pH 8.0) containing 1 mM PMSF and 1× protease inhibitor (MCE) was added into the powder to promptly homogenize the powder. After centrifugation ($14,000 \times g$, 10 min), 80 μL supernatant was mixed with 20 μL 5×SDS loading buffer (250 mM Tris-HCl pH 6.8, 25% glycerol, 10% SDS, 0.01% Bromo Phenol Blue, 10 mM DTT, 5% β-mercaptoethanol), boiled at 95 °C for 5 min, and analyzed by western blot immediately or stored at −20 °C until further application. Quantification of immunoblots was performed using ImageJ.

## Co-immunoprecipitation (Co-IP) assays

Seven-day-old WLc-grown Col, *35S: GFP-TCP15*, *35S: GFP-TCP21*, *35S: HA-LAO1*, or *35S: GFP-DSK2B/35S: HA-LAO1* seedlings were used. The whole seedlings were grounded into fine powder in liquid nitrogen, and 1 mL IP buffer (50 mM Tris-HCl pH 7.5, 150 mM NaCl, 1 mM $MgCl_2$, 0.1% NP40) containing 1 mM PMSF and 1×protease inhibitor was added. After 3 times of centrifugation ($12,000 \times g$, 5 min) at 4 °C, the supernatant was mixed with 20 μL anti-HA or anti-GFP Affinity Beads (Smart-lifesciences) and rotated at 4 °C for 2 h to enrich the bait proteins. The pellets were washed with IP buffer for at least 3 times, and boiled in 2×SDS loading buffer at 95 °C for 5 min to elute proteins. The protein samples were stored at −20 °C or analyzed by western blot immediately.

## Recombinant protein expression and purification

The Rosetta (DE3) strain harboring corresponding construct was cultured until OD reaches 0.5, and then a final concentration of 0.5 mM IPTG was added to induce the expression of recombinant protein at 16 °C for 20 h. After sonication and centrifugation, and the recombinant protein was purified with Ni-NTA agarose (Sangon) and GST agarose (Yesen) for His-tagged and GST-tagged protein, respectively, under manufacturer's instructions. Glycerol was added into the purified recombinant protein to a final concentration of 10%, and then the protein was aliquoted and stored at −80 °C.

## GST pull-down assays

For GST-pull-down assays, GST or each GST-tagged recombinant proteins were incubated with indicated His-tagged proteins in 1 mL GST-pull-down buffer (50 mM Tris-HCl pH 7.5, 150 mM NaCl, 0.5% Trition X-100 and 1 mM PMSF) at 4 °C for 3 h on a tube rotator. Following incubation, 20 μL GST beads were added and the incubation was continued for an additional 1 h. The pellets were washed with GST-pull-down buffer for at least 3 times, and then boiled in 2×SDS loading buffer at 95 °C for 5 min to elute proteins for western blot analysis.

## Semi-in vivo pull-down assays

Seven-day-old Col and *35S: HA-LAO1* seedlings were grounded into powder in liquid nitrogen and homogenized in IP buffer (50 mM Tris-HCl pH 7.5, 150 mM NaCl, 1 mM $MgCl_2$, 0.1% NP40) containing 1 mM PMSF and 1×protease inhibitor. After centrifugation ($12,000 \times g$, 5 min) three times at 4 °C, the supernatant was mixed with 20 μL anti-HA Affinity Beads (Smart-lifesciences) and rotated at 4 °C for 1 h to enrich HA-LAO1 proteins. After 3 times of wash with IP buffer, the beads were mixed with 1 μg of His-DSK2B or His-TCP14 purified proteins for 3 h at 4 °C. The pellets were washed with IP buffer for at least 3 times, and then boiled in 2×SDS loading buffer at 95 °C for 5 min to elute proteins for western blot analysis.

## GUS staining

The *proLAO1(3k): GUS* and *proLAO1(2k): GUS* transgenic plants were treated with nitrogen deficiency, and untreated plants were used as control for subsequent staining process with the GUS staining kit (Coolaber). Finally, the stained seedlings were destained with 70% ethanol for 2–3 times, and the seedlings were imaged with MICROTEK ScanMaker.

## Protoplast isolation and transfection

Protoplasts were isolated and transfected as previously described, with some minor modifications (Dong et al, 2020; Yoo et al, 2007). Generally, the protoplasts were transfected with 20–30 μg of the indicated plasmids for up to 15 min, and the protoplasts were incubated in darkness at 22 °C overnight before downstream applications were performed. For nitrate feeding assay of *Arabidopsis* protoplast, the protoplasts were incubated in 1 mL of incubation solution (0.2 M mannitol, 20 mM MES, 20 mM KCl, 0.2 M $CaCl_2$) with or without 5 mM $KNO_3$ (use KCl at the corresponding concentration as the control) at 22 °C overnight. Protoplasts were collected by centrifugation for 2 min. The LUC and REN activities were measured using Dual-luciferase reporter assay kit (Vazyme) and the signals were captured by Lumipro (Promega).

## Bimolecular fluorescence complementation

The full-length coding sequence of *LAO1* or those of *TCP14* and *TCP21* were cloned in-frame of the sequence encoding the N-terminal half of YFP (YN) or the C-terminal half of YFP (YC), respectively. Each construct was then transformed into agrobacteria GV3101. *LAO1-YN* was co-infiltrated together with *TCP14-YC* or *TCP21-YC* into the leaves of 1-month-old *N.benthamiana* plants, along with appropriate controls. YFP signal was observed after 48 h of infiltration using a laser scanning confocal microscope. The primers used in the study are listed in Appendix Table S1.

## Generation of DSK2 polyclonal antibody

Anti-DSK2 antibody was produced using recombinant full-length DSK2B protein bearing a N-terminal 6×His tag. Rabbits were immunized with purified His-DSK2B full-length recombinant protein. Primary immunization was followed by booster injections every 2–4 weeks. Polyclonal antibodies were purified from immunized serum using Protein A/G affinity chromatography (Beijing Protein Innovation).

## Accession numbers

We used existing sequence data available in The *Arabidopsis* Information Resource (TAIR). The sequence data in this study are available in TAIR under the following accession numbers: *LAO1* (AT1G67480), *DSK2B* (AT2G17200), *DSK2A* (AT2G17190), *ATG5* (AT5G17290), *ATG7* (AT5G45900), *TCP7* (AT5G23280), *TCP14* (AT3G47620), *TCP15* (AT1G69690), *TCP21* (AT5G08330), *NIA2* (AT1G37130), *ACT7* (AT5G09810).

# Data availability

This study did not generate any sequencing data that needed to be deposited in public databases.

The source data of this paper are collected in the following database record: biostudies:S-SCDT-10_1038-S44319-025-00491-9.

# Peer review information

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

## Acknowledgements

We thank Dr. Zeng Tao for help with the Y2H screening experiment. We thank Dr. Ronghui Pan for *atg5-1* and *atg7-3* mutant seeds, Dr. Wei Su and Dr. Benke Kuai for *tcp14-2* and *tcp15-1* mutant seeds, Dr. Yunhai Li for *tcp14-3 tcp15-3* mutant seeds, and Dr. Sujuan Cui for *tcpS* mutant seeds. We thank Ms. Yunqin Li for technical support to confocal microscopy observations. This work was supported by National Natural Science Foundation of China (32470286; 32100190), the National Natural Science Fund for Excellent Young Scientists Fund Program (Overseas), and the Zhejiang University Global Partnership Fund (100000-11320).

## Author contributions

**Yuanyuan Li**: Data curation; Formal analysis; Validation; Investigation; Visualization; Writing—original draft. **Shuyang Cheng**: Data curation; Investigation. **Xu Jin**: Resources; Data curation; Investigation. **Ruoxuan Wu**: Resources; Data curation; Investigation. **Yiyi Guo**: Resources; Data curation; Investigation; Visualization. **Dezhi Wu**: Resources; Writing—review and editing. **Jie Dong**: Conceptualization; Resources; Supervision; Funding acquisition; Project administration; Writing—review and editing.

Source data underlying figure panels in this paper may have individual authorship assigned. Where available, figure panel/source data authorship is listed in the following database record: biostudies:S-SCDT-10_1038-S44319-025-00491-9.

## Disclosure and competing interests statement

The authors declare no competing interests.

# Expanded View Figures

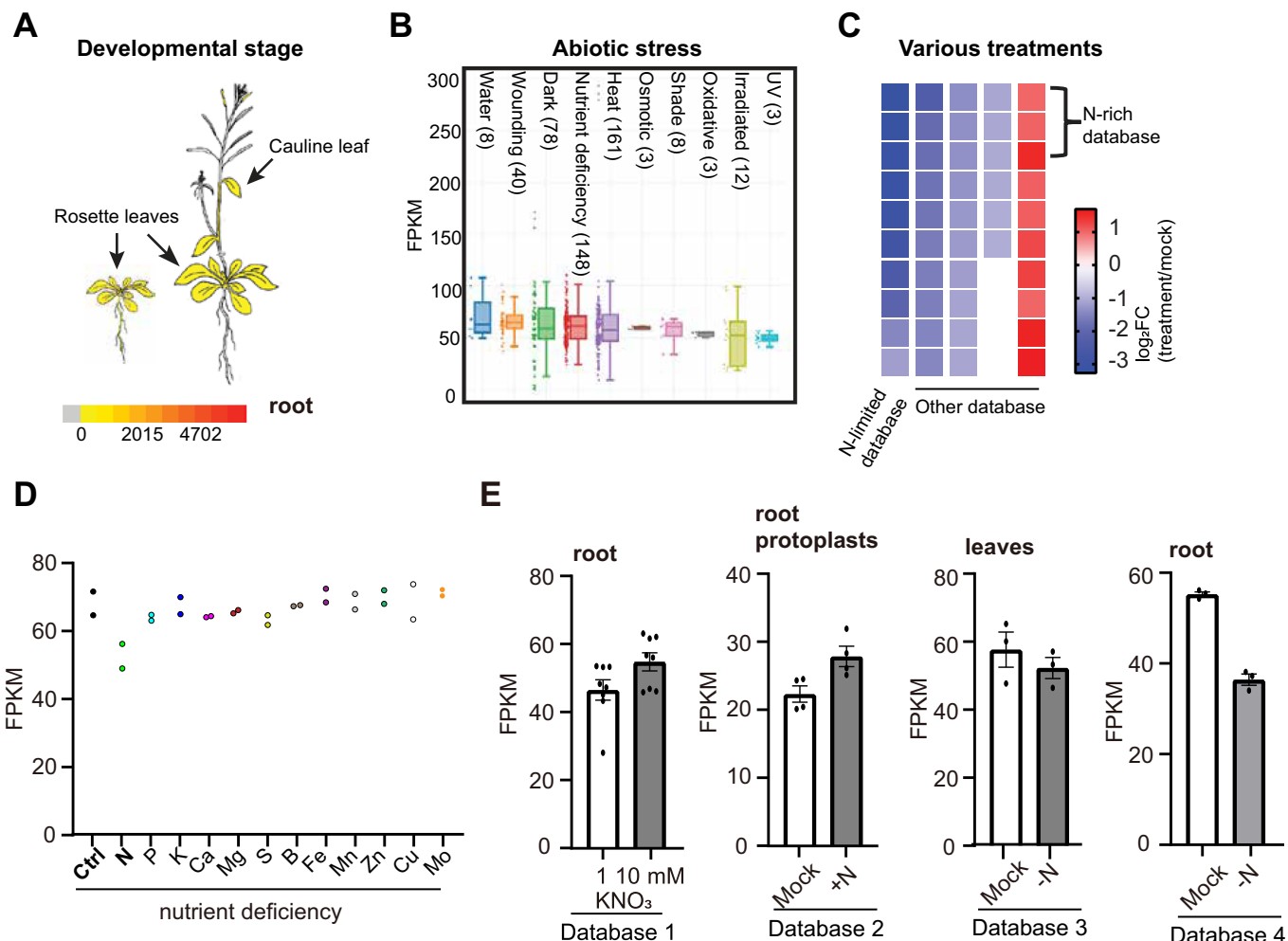

**Figure EV1. *LAO1* is a nitrogen-responsive gene.**

(A) *LAO1* predominantly expressed in rosette and cauline leaves. Data were retrieved from *Arabidopsis* eFP browser. (B) The expression level of *LAO1* (as determined by the mean FPKM of *LAO1* under the conditions in each category) among top 10 abiotic stress conditions. FPKM, Fragments Per Kilobase of transcript per Million mapped reads. The graph was generated using online tools (https://plantrnadb.com/athrdb/). (C–E) *LAO1* expression level was tightly associated with nitrogen availability. (C) A heatmap representation of the expression levels of *LAO1* in 46 RNA-seq databases from various treatments. Each square represents the $\log_2FC$ (treatment/mock) of *LAO1* in each treatment. (D) Deficiency of nitrogen, but not other nutrients, downregulated the expression of *LAO1* gene. Data were shown as individual datapoints of two independent replicates. (E) Elevated nitrogen level promoted *LAO1* expression in both root and root protoplasts (Databases 1&2), while nitrogen starvation inhibited *LAO1* expression in both leaves and root (Databases 3&4). Data were shown in mean ± SEM of at least 3 biological replicates. The databases used for gene expression analyses of LAO1 in (C), (D), and (E) were listed in Dataset EV1 file.

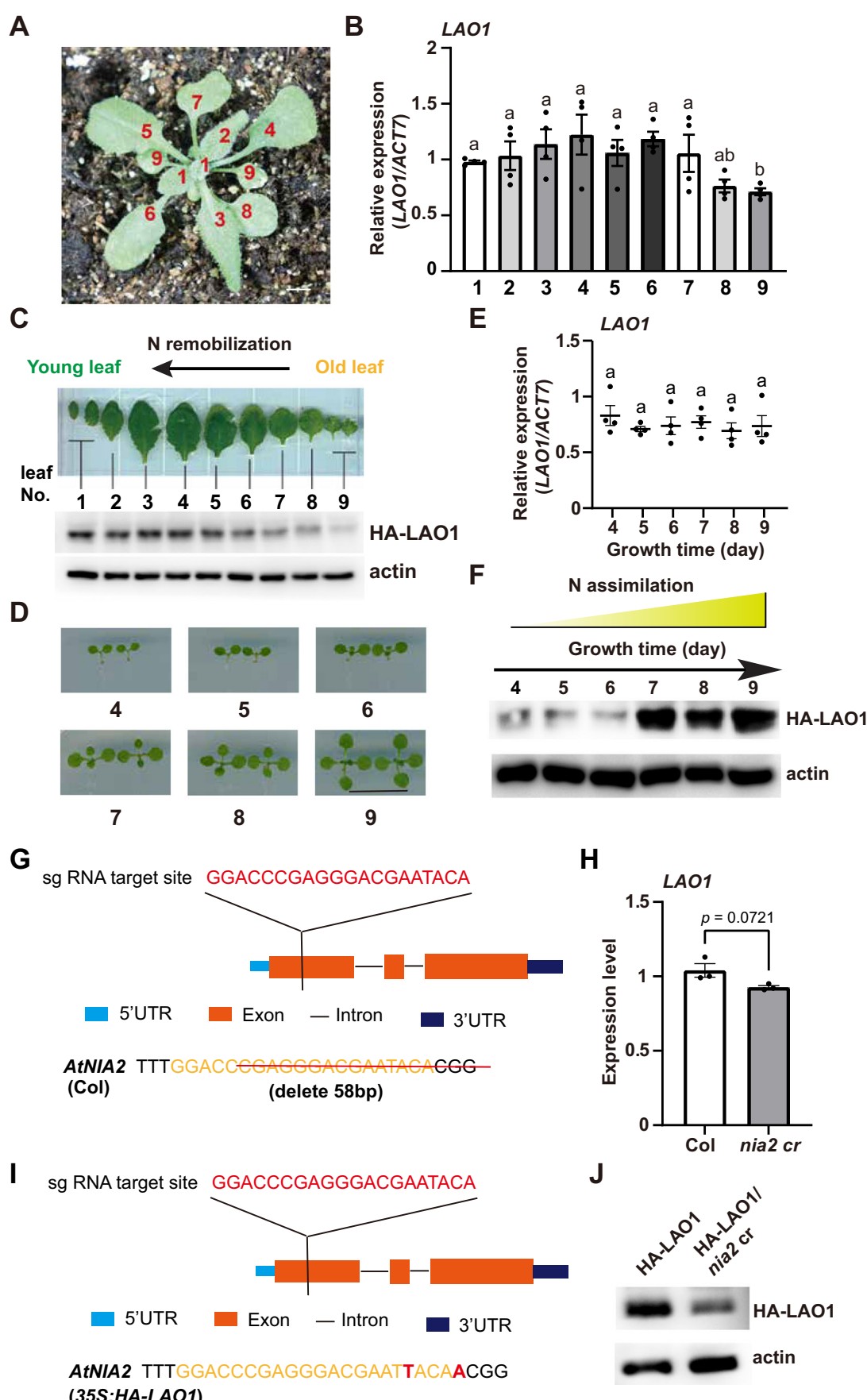

**Figure EV2. LAO1 abundance is associated with nitrogen dynamics in plants.**

(A) A representative graph illustrates the numbering of leaves on an adult plant for downstream applications. The two youngest leaves were grouped together as 1, while the two oldest leaves were grouped together as 9. Scale bar: 1 cm. (B) The relative expression levels of *LAO1* in different leaves of 3-week-old Col adult plant. *ACT7* was used as internal control. Data were shown in mean ± SEM of 4 different plants. (C) The protein levels of HA-LAO1 in different leaves of 3-week-old *35S: HA-LAO1* adult plant. Total proteins were extracted from similar amounts of leaves with the indicated number, and western blots were performed using anti-HA and anti-actin antibodies. (D) A representative graph showing the morphology of seedling grown under continuous white light condition in a growth chamber at 22 °C for 4 to 9 days after stratification (DAS). Scale bar: 1 cm. (E) The relative expression levels of *LAO1* in Col seedlings from 4 to 9 DAS. *ACT7* was used as internal control. Data were shown in mean ± SEM of 4 biological replicates. (F) The protein levels of HA-LAO1 in *35S: HA-LAO1* seedlings, from 4 to 9 DAS. Total proteins were extracted from *35S: HA-LAO1* seedlings at the indicated growth times. Western blots were performed using anti-HA and anti-actin antibodies. (G, I) Mutation induced by CRISPR-Cas9 at *NIA2* gene locus. The gene structure of *NIA2* and the position of sgRNA were shown. Mutation was confirmed by PCR-sequencing of targeted locus in both Col (G) and *35S: HA-LAO1* transgenic background (I). (H, J) Loss-of-function mutations of NIA2 did not affect *LAO1* mRNA level but downregulated HA-LAO1 protein level. (H) The relative expression levels of *LAO1* in 7-day-old Col and *nia2 cr* mutant seedlings. *ACT7* was used as internal control. Data were shown in mean ± SEM of 3 biological replicates. (J) Immunoblot analyses of HA-LAO1 protein levels in 7-day-old *35S: HA-LAO1* and *35S: HA-LAO1/nia2 cr* seedlings. Western blot was performed using anti-HA and anti-actin antibodies. In (B) and (E), statistical significance was determined using one-way ANOVA analysis ($p < 0.05$, ANOVA followed by Tukey's post hoc comparison test). Different letters denoted significant differences. In (H), statistical significance was calculated by Student's $t$ test.

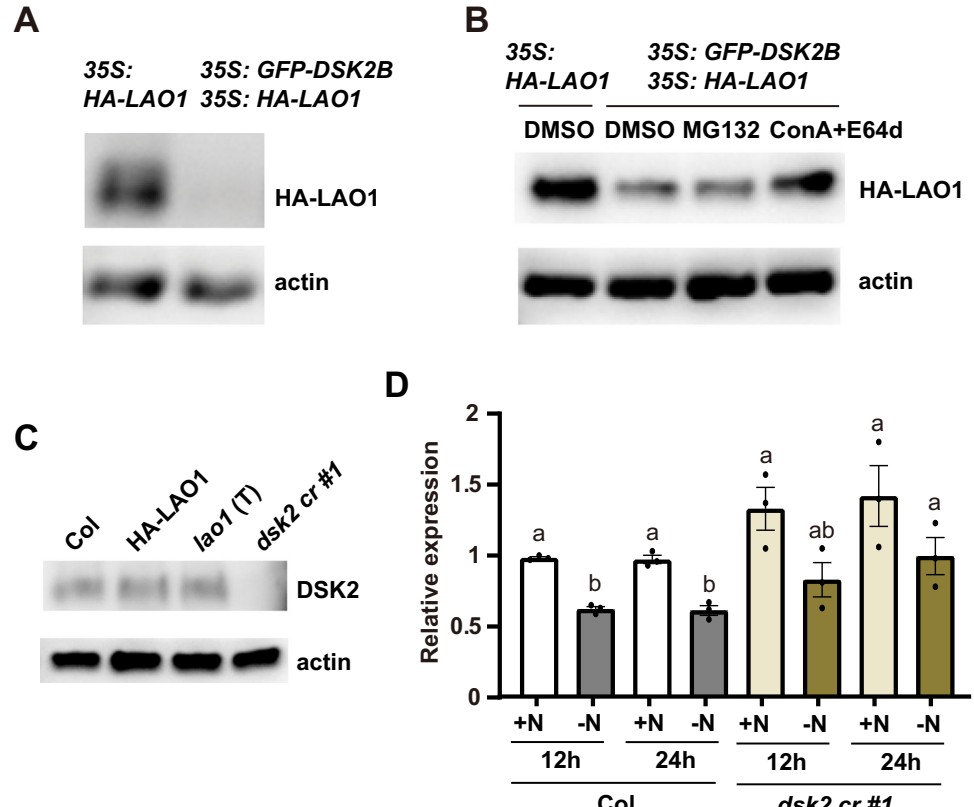

**Figure EV3.   Analysis of the mutual regulation relationship between DSK2 and LAO1.**

(**A**) Overexpression of *DSK2B* reduced HA-LAO1 protein level. Total proteins were extracted from 7-day-old *35S: HA-LAO1* and 35S: *GFP-DSK2B/35S: HA-LAO1* seedlings, and analyzed by immunoblot using anti-HA and anti-actin antibodies. (**B**) ConA and E64d treatment but not MG132 inhibited *DSK2* overexpression induced degradation of HA-LAO1 protein. Seven-day-old *35S: HA-LAO1* and *35S: GFP-DSK2B/35S: HA-LAO1* seedlings were treated with DMSO (control), 50 μM MG132, or 1 μM ConA and 50 μM E64d for 4 h. Total proteins were extracted and analyzed by immunoblot using anti-HA and anti-actin antibodies. (**C**) LAO1 does not affect the protein level of DSK2. Seven-day-old Col, *HA-LAO1*, *lao1*(T), and *dsk2 cr* seedlings were harvested. Total proteins were extracted and analyzed by immunoblot using anti-DSK2 and anti-actin antibodies. (**D**) RT-qPCR analysis of *LAO1* in the *dsk2* cr mutant following nitrogen starvation treatment. *ACT7* was used as internal control. Data were shown in mean ± SEM of 3 biological replicates. Statistical significances were determined using one-way ANOVA analysis ($p < 0.05$, ANOVA followed by Tukey's post hoc comparison test). Different letters denoted significant differences.

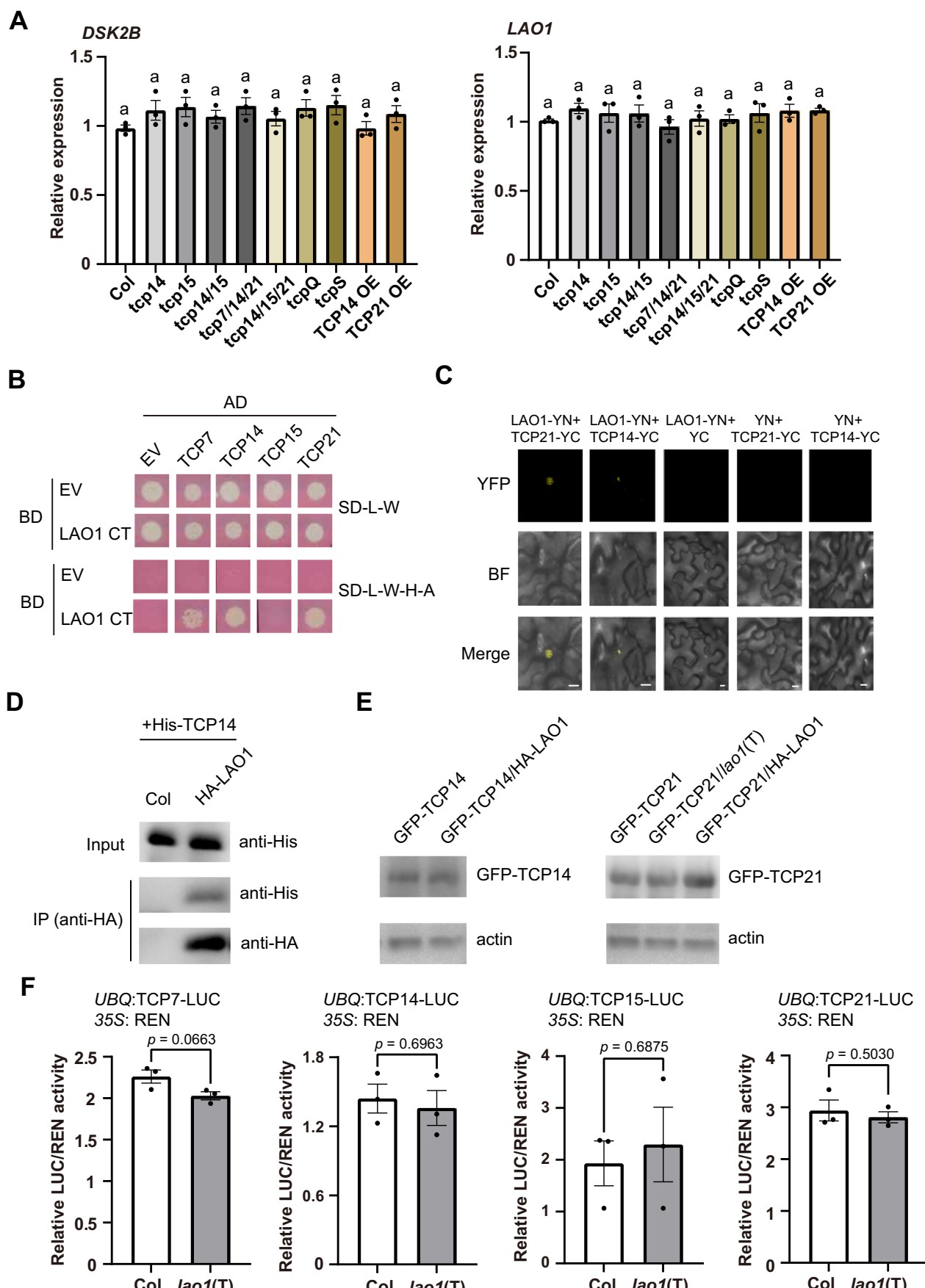

◀ **Figure EV4. Gene expression analyses of *DSK2* and *LAO1* in *TCP* genetic materials, along with regulatory analyses between LAO1 and TCPs.**

(A) TCPs do not affect the expression of *DSK2B* or *LAO1*. *ACT7* was used as internal control. Data were shown in mean ± SEM of 3 biological replicates. The statistical significances were determined using one-way ANOVA analysis ($p < 0.05$, ANOVA followed by Tukey's post hoc comparison test). Different letters denoted significant differences. (B) LAO1 interacted with TCPs in yeast. Corresponding constructs were co-transformed into the yeast strain Y2H-gold. The colonies appeared on SD-L-W plates were transferred onto SD-L-W-H and SD-L-W-A-H plates for interaction test. (C) BiFC assay in leaves of one-month-old *N.benthamiana* plants using Agrobacteria GV3101 mediated transient expression. Scale bars: 20 μm. (D) HA-LAO1 interacted with recombinant His-TCP14 in semi-in vivo pull-down assays. HA-LAO1 proteins were immunoprecipitated from transgenic *35S: HA-LAO1* plants, and then incubated with recombinant His-TCP14 protein. The pellets were boiled in 2×SDS loading buffer, and the samples were detected by western blotting using anti-His and anti-HA antibodies. (E) Immunoblot analyses of GFP-TCP14 or GFP-TCP21 protein abundance. Total proteins were extracted from the leaves of 7-day-old *GFP-TCP14*, *GFP-TCP14/HA-LAO1*, *GFP-TCP21*, *GFP-TCP21/HA-LAO1*, and *GFP-TCP14/lao1*(T) plants. Western blots were performed using anti-GFP and anti-actin antibodies. (F) The luminescence ratio of TCP-LUC/REN in Col and *lao1*(T) mutant protoplasts. Experimental design was similar as Fig. 6E. Relative TCP-LUC/REN activity was shown in mean ± SEM of 3 independent experiments. All the statistical significances were calculated by the Student's *t* test.

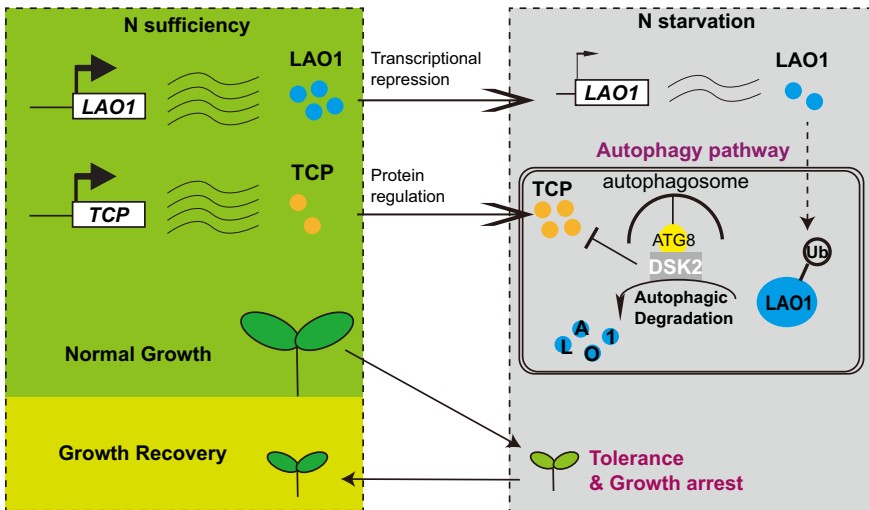

**Figure EV5.   A simplified model for deciphering the role of DSK2-LAO1-TCP module in plant response to nitrogen starvation.**

Under N-sufficient conditions, active N assimilation maintains high levels of LAO1, while relatively lower levels of TCP proteins, supporting optimal growth. Upon N starvation, LAO1 protein is degraded through DSK2-mediated autophagy, whereas TCP proteins accumulate, enhancing plant tolerance to nitrogen starvation. Concurrently, DSK2 modulates TCP protein stability to balance growth and tolerance.

