## [Peer Review File · EMBO Reports]

DSK2-mediated degradation of F-box protein LAO1 and class I TCPs modulates the nitrogen starvation response

Yuanyuan Li, Shuyang Cheng, Xu Jin, Ruoxuan Wu, Yiyi Guo, Dezhi Wu, and Jie Dong

Corresponding author(s): Jie Dong (jie_dong@zju.edu.cn)

Review Timeline:

Submission Date:	29th Aug 24
Editorial Decision:	26th Sep 24
Revision Received:	26th Dec 24
Editorial Decision:	13th Feb 25
Revision Received:	11th Mar 25
Editorial Decision:	16th Apr 25
Revision Received:	30th Apr 25
Accepted:	8th May 25

Transaction Report:

Dear Dr. Dong

Thank you for the submission of your research manuscript to our journal. We have now received the full set of referee reports that is copied below.

As you will see, the referees acknowledge that the findings are interesting and that the conclusions are overall supported by the data presented but they also raise a number of concerns and have suggestions how to further strengthen the data that need to be addressed. I agree with referee #1 and #3 that the involvement of LAO1 in drought tolerance has not been investigated in depth and would concur with the suggestion to remove the drought aspect from the manuscript.

Given the constructive comments and overall positive evaluation, we would like to invite you to revise your manuscript with the understanding that the referee concerns (as detailed above and in their reports) must be fully addressed and their suggestions taken on board. Please address all referee concerns in a complete point-by-point response. Acceptance of the manuscript will depend on a positive outcome of a second round of review. It is EMBO Reports policy to allow a single round of revision only and acceptance or rejection of the manuscript will therefore depend on the completeness of your responses included in the next, final version of the manuscript.

We realize that it is difficult to revise to a specific deadline. In the interest of protecting the conceptual advance provided by the work, we recommend a revision within 3 months (December 30). Please discuss the revision progress ahead of this time with the editor if you require more time to complete the revisions.

I am also happy to discuss the revision further via e-mail or a video call, if you wish.

****IMPORTANT NOTE:

We perform an initial quality control of all revised manuscripts before re-review. Your manuscript will FAIL this control and the handling will be delayed IN CASE the following APPLIES:

- 1) A data availability section providing access to data deposited in public databases is missing. If you have not deposited any data, please add a sentence to the data availability section that explains that.
- 2) Your manuscript contains statistics and error bars based on $n=2$. Please use scatter blots in these cases. No statistics should be calculated if $n=2$.

When submitting your revised manuscript, please carefully review the instructions that follow below. Failure to include requested items will delay the evaluation of your revision. *****

- 1) a .docx formatted version of the manuscript text (including legends for main figures, EV figures and tables). Please make sure that the changes are highlighted to be clearly visible.
- 2) individual production quality figure files as .eps, .tif, .jpg (one file per figure). Please download our Figure Preparation Guidelines (figure preparation pdf) from our Author Guidelines pages <https://www.embopress.org/page/journal/14693178/authorguide> for more info on how to prepare your figures.
- 3) a .docx formatted letter INCLUDING the reviewers' reports and your detailed point-by-point responses to their comments. As part of the EMBO Press transparent editorial process, the point-by-point response is part of the Review Process File (RPF), which will be published alongside your paper.
- 4) a complete author checklist, which you can download from our author guidelines (<<https://www.embopress.org/page/journal/14693178/authorguide>>). Please insert information in the checklist that is also reflected in the manuscript. The completed author checklist will also be part of the RPF.
- 5) Please note that all corresponding authors are required to supply an ORCID ID for their name upon submission of a revised manuscript (<<https://orcid.org/>>). Please find instructions on how to link your ORCID ID to your account in our manuscript tracking system in our Author guidelines (<<https://www.embopress.org/page/journal/14693178/authorguide#authorshipguidelines>>)

6) We replaced Supplementary Information with Expanded View (EV) Figures and Tables that are collapsible/expandable online. A maximum of 5 EV Figures can be typeset. EV Figures should be cited as "Figure EV1, Figure EV2" etc... in the text and their respective legends should be included in the main text after the legends of regular figures.

7) Primary datasets (and computer code, where appropriate) produced in this study need to be deposited in an appropriate public database (see < <https://www.embopress.org/page/journal/14693178/authorguide#dataavailability>>).

The accession numbers and database should be listed in a formal "Data Availability " section (placed after Materials & Method) that follows the model below (see also < <https://www.embopress.org/page/journal/14693178/authorguide#dataavailability>>). Please note that the Data Availability Section is restricted to new primary data that are part of this study.

Data availability

Additional information on source data and instruction on how to label the files are available <<https://www.embopress.org/page/journal/14693178/authorguide#sourcedata>>.

10) Figure legends and data quantification:

- the name of the statistical test used to generate error bars and P values,
- the number (n) of independent experiments (please specify technical or biological replicates) underlying each data point,
- the nature of the bars and error bars (s.d., s.e.m.)
- If the data are obtained from n {less than or equal to} 5, show the individual data points in addition to the SD or SEM.
- If the data are obtained from n {less than or equal to} 2, use scatter blots showing the individual data points.

11) Our journal encourages inclusion of *data citations in the reference list* to directly cite datasets that were re-used and obtained from public databases. Data citations in the article text are distinct from normal bibliographical citations and should directly link to the database records from which the data can be accessed. In the main text, data citations are formatted as follows: "Data ref: Smith et al, 2001" or "Data ref: NCBI Sequence Read Archive PRJNA342805, 2017". In the Reference list,

data citations must be labeled with "[DATASET]". A data reference must provide the database name, accession number/identifiers and a resolvable link to the landing page from which the data can be accessed at the end of the reference. Further instructions are available at <<https://www.embopress.org/page/journal/14693178/authorguide#referencesformat>>.

12) All Materials and Methods need to be described in the main text using our 'Structured Methods' format. According to this format, the Methods section includes a Reagents and Tools Table (listing key reagents, experimental models, software and relevant equipment and including their sources and relevant identifiers) followed by a Methods and Protocols section describing the methods, ideally using a step-by-step protocol format. The aim is to facilitate adoption of the methodologies across labs. Please download and fill our Reagents and Tools Table template (.docx), which you can find in our author guidelines:

13) As part of the EMBO publication's Transparent Editorial Process, EMBO Reports publishes online a Review Process File to accompany accepted manuscripts. This File will be published in conjunction with your paper and will include the referee reports, your point-by-point response and all pertinent correspondence relating to the manuscript.

Yours sincerely,

=====

Referee #1:

This is a very solid and highly informative manuscript reporting an interesting new link between autophagy and the ubiquitination pathway. It reads very well, and the experiments are well thought of. In my opinion however the part with the TCP interactors and their drought phenotypes still needs to be further elaborated and I would suggest to omit it at this stage, while strengthening instead the connection between LAO1 and DSK2, as suggested in my comments below.

-I suggest to carry out ANOVA statistical test and use letter grouping in the graphs, whenever possible, similar to those used in Figure 7. This might also help strengthening the overall statistical relevance of the results, which for some of the data -especially the mutant phenotypes (see for example figure 1 E-G and 2C)- is rather low.

The downregulation of LAO1 expression after 48 and 96 hours (72 hours for GUS) on -N conditions seem to indicate that it is not directly regulated by the N context. Do the authors have any comment on that? Also, The LAO-GUS 2k vs 3k expression is rather different, do the authors have any explanation for this?

Lines 139 -140. it might be easier for the reader to provide a bit more background on how LAO1 was discovered (proteomics experiment etc).

Figure 3A, can the authors discuss why the effect of the autophagy inhibitor seem transient (it almost disappears at 1 h treatment). Using a densitometric analysis might help understanding if this is reproducible or significant.

Figure 3B, there is still a bit of HA-LAO degradation even in the atg mutants, can the authors discuss this?

Figure 4A a similar area of the gel blot should also be shown for the anti-HA immunoblot

It is interesting that while the difference in LAO1 levels in the *dsk2* background compared to the wild type is very evident in Figure 4E, it is less evident in the CHX chase (Figure 4G), and it is also different from the LAO1 levels in the CHX case shown in Figure 3A and B, Can the authors discuss it?

Supplemental Figure 2. Can the authors describe more in detail the databases that were used?

Suppl fig 4. There are two #1 on figure 4A. can the authors explain?

Supplemental figure 6, the seedlings are rather small in the photos, can the authors increase the magnification and the resolution of the images shown?

Referee #2:

Li et al., identified a novel module that involves an F-box protein LAO1, a ubiquitin receptor protein DSK2, and class I TCP transcription factors regulating Arabidopsis response to nitrogen deficiency and drought tolerance. While the paper is interesting, well-written, and generally robust in its experimental approach, some additional experimental validation or clarification of the conclusions is necessary to strengthen it further and to clarify the link between the regulation of drought and nitrogen stress responses by the identified regulatory module

1) I suggest that the authors include a schematic model for visualizing their findings.

2) I am missing any direct evidence that LAO1 can be recruited to autophagosomes via DSK2 and degraded by autophagy. The authors could assess this using several relatively straightforward assays, especially since many of the necessary transgenic lines are already available in their labs.

- Assess co-localization of mCherry-LAO1 and GFP-ATG8 in both Col-0 and *dsk2* cr mutant background under control/ nitrogen starvation conditions (using ConA treatment)
- Assess co-localization of mCherry-LAO1 and GFP-DSK2 under control/ nitrogen starvation conditions (using ConA treatment)
- Detect the transport of the tagged LAO1 to the vacuole under control/nitrogen starvation (using ConA treatment)

3) Mutations in DSK2 lead to better performances in both Col and HA-LAO1 backgrounds after nitrogen starvation (Figure 5E). This is in part due to the link between DSK2 and TCPs. However, could the author rule out the possibility of the reduced expression of LAO1 (and therefore its protein level) in the *dsk2* cr mutants? (e.g., in case DSK2 controls the transcriptional regulator of LAO1 as an alternative regulatory mechanism). For a comprehensive assessment, the authors should measure the transcript abundance of LAO1 in *dsk2* cr mutants (under control and nitrogen deficiency conditions).

4) How does DSK2 mediate the degradation of TCPs? Is there evidence to include or rule out the involvement of autophagy in this process?

5) Does the DSK2-mediated degradation of TCPs occur during nitrogen starvation as well?

6) Is the effect of TCPs mitigating LAO1 specific to N starvation, or does it also apply to drought? To address this, the authors could test the phenotypes of HA-LAO1/GFP-TCP14/ and HA-LAO1/GFP-TCP21 in response to drought conditions.

7) Please show the phenotype of 35S: HA-LAO/*atg5* (and *atg7-3*) mutant analyzed in Figure 3C, especially after 24h of N deficiency. I assume these plants barely survive the stress.

Minor comments:

1. Figure 3A: the immunoblotting data should be quantified, especially after 60 min.

2. Line 167 describes the experimental setup as "followed by a two-day recovery on 1/2 MS plates," while in the legends of Figures 1D& E, a four-day recovery is mentioned. Please clarify this inconsistency.

3. Figure 7B: It is unclear which -N condition presented in panel A is quantified in B? Is it -N 6d, R5d or -N 7d, R4d?

Col-0: I suggest using the standard format, 'Col-0,' rather than just 'Col.' Or specify 'Col-0' the first time it is mentioned in the main text to indicate which Columbia accession is being referred to.

Referee #3:

The manuscript presents a considerable amount of working looking at the potential role of the F-box protein LAO1, the pexophagy receptor DSK2 that binds to and target proteins for autophagic degradation, and transcription factors from the TCP family, in responses to nitrogen limitation and drought stress.

The title of the manuscript is not supported by the results presented. The connection between the proteins of interest, nutrient stress and drought stress is not made. Analysis of involvement in drought stress response is very limited, with no results shown for LAO1 involvement in drought tolerance.

What is meant by -N? The material and method section does not include information on N concentrations in growth media. This is a significant omission. Does the media completely lack N or is N limited? If limited, is it mild or severe N limitation? As the authors may be aware, plants respond differently depending on the severity of N limitation (see *Plant physiology* 166(2) 509-517; *Molecular Plant*, 15(1), 86-103). What is the source of N?

Figure 2C - While the level of LAO1 is lower under -N compared to +N at each timepoint, the level of LAO1 seems to be increasing over time under both +N and -N conditions.

The degradation rate of LAO1 under -N needs to be quantified and compared to that under +N (Figure 3A-C). To evaluate the role of DSK2 in modulating LAO1 abundance, CHX assays should be used to examine the degradation of LAO1 in the presence and absence of DSK2 under +N and -N conditions.

Figure 5E, loss of DSK2 should lead to the overaccumulation/high levels of LAO1. - should the *dsk2* mutant not look similar to the LAO1 overexpressor (OE) ? And should the *dsk2* LAO1 OE be more sensitive to -N conditions compare to Col-0 and LAO1 OE? Quantification is needed for the seedlings in Figure 5E. Similar experiments are required for examining the impact of DSK2 on TCP stability.

Does TCP regulate the expression of LAO1 and/or DSK2?

We sincerely thank all the reviewers for their critical and constructive comments which have significantly improved the manuscript. We have responded to the comments point by point below, and hope our responses will meet with their full approval.

Referee #1:

This is a very solid and highly informative manuscript reporting an interesting new link between autophagy and the ubiquitination pathway. It reads very well, and the experiments are well thought of. In my opinion however the part with the TCP interactors and their drought phenotypes still needs to be further elaborated and I would suggest to omit it at this stage, while strengthening instead the connection between LAO1 and DSK2, as suggested in my comments below.

Author Reply: We thank the reviewer for the positive evaluation of our study and for the critical comments. As suggested, we have removed the drought phenotypes from the revised manuscript.

-I suggest to carry out ANOVA statistical test and use letter grouping in the graphs, whenever possible, similar to those used in Figure 7. This might also help strengthening the overall statistical relevance of the results, which for some of the data -especially the mutant phenotypes (see for example figure 1 E-G and 2C)- is rather low.

Author Reply: We appreciate the reviewer's valuable suggestion. In the revised manuscript, we have conducted ANOVA statistical tests for the data presented in the following figures: Figure 1E, Figure 1G, Figure 3 A-C, Figure 3F, Figure 3H, Figure 4F, Figure 4G, Figure 5B, Figure 5F, Figure 6F, Figure 6G, Figure 7B, Figure 7D, Figure 7E, Figure EV2B, Figure EV2E, and Figure EV3D.

The downregulation of LAO1 expression after 48 and 96 hours (72 hours for GUS) on -N

conditions seem to indicate that it is not directly regulated by the N context. Do the authors have any comment on that? Also, The LAO-GUS 2k vs 3k expression is rather different, do the authors have any explanation for this?

Author Reply: Thank you for your remarks. To clarify this, we performed a time-course gene analysis of *LAO1* in the revised manuscript. As shown in Figure 2A, the downregulation of *LAO1* mRNA levels was evident as early as 12 hours after nitrogen (N) depletion. This result suggests that *LAO1* gene expression is likely regulated directly by N starvation, though the detailed mechanism requires further investigation.

For GUS staining, we provided additional lines for both proLAO1(3k): GUS and proLAO1(2k): GUS. In all four different lines, N depletion significantly decreased GUS signal, supporting the conclusion that N starvation downregulates *LAO1* expression. However, we noticed that the GUS signals of proLAO1(2k): GUS lines were relatively weaker than those of proLAO1(3k): GUS lines under both +N and -N conditions. This expression difference may arise from the different regulatory elements present in the proLAO1(3k) and proLAO1(2k) promoters, which requires further investigation. We have discussed about this in the discussion part of the revised manuscript. Lines 440-445

Lines 139 -140. it might be easier for the reader to provide a bit more background on how *LAO1* was discovered (proteomics experiment etc).

Author Reply: As suggested, we have provided more details on how we identified and focused on *LAO1* through an immunoprecipitation coupled with mass spectrometry (IP-MS) using *CUL1* as bait in the main text of the revised manuscript. Lines 135-140.

Figure 3A, can the authors discuss why the effect of the autophagy inhibitor seem transient (it almost disappears at 1 h treatment). Using a densitometric analysis might help understanding if this is reproducible or significant.

Author Reply: We repeated and quantified this experiment. The new results in the revised Figure 3A suggest that the effect of E64d treatment was, somehow, more obvious and significant at 30 min as compared to 60 min. One possible explanation for this phenomenon is that there might be other unknown mechanisms, such as peptidase-mediated proteolysis, that control the protein turnover of *LAO1* after N starvation. We have discussed this result in more detail in the revised manuscript.

Lines 475-478

Figure 3B, there is still a bit of HA-LAO degradation even in the *atg* mutants, can the authors discuss this?

Author Reply: Thank you for pointing this out. We agree with the reviewer that HA-LAO1 still undergoes degradation in *atg* mutant backgrounds under N starvation. We believe this observation is related to the transient effect of E64d treatment mentioned above. The results together suggest that there are additional mechanisms controlling LAO1 protein proteolysis. We have discussed these findings in detail in the revised manuscript. Lines 475-478

Figure 4A a similar area of the gel blot should also be shown for the anti-HA immunoblot

Author Reply: As suggested by the reviewer, we have included a similar area of the gel blot for anti-HA immunoblot in the revised Figure 4A.

It is interesting that while the difference in LAO1 levels in the *dsk2* background compared to the wild type is very evident in Figure 4E, it is less evident in the CHX chase (Figure 4G), and it is also different from the LAO1 levels in the CHX case shown in Figure 3A and B,

Can the authors discuss it?

Author Reply: Thank you for bringing this issue to our attention. When performing western blots, we typically reuse diluted antibodies multiple times. Occasionally, this practice can affect the appearance and intensity of the bands, which might explain the discrepancy in the previous manuscript. In the revised manuscript, we have repeated the relevant experiments using freshly diluted antibodies and quantified the results, as shown in Figure 4F and Figure 4G. We believe that the differences in the updated results are now evident, and the bands are consistent with those shown in Figure 3A and 3B.

Supplemental Figure 2. Can the authors describe more in detail the databases that were used?

Author Reply: Thank you for your suggestion. Both databases are public online

platforms for gene expression profile analyses of Arabidopsis genes. As suggested, we have updated the website for 'The Arabidopsis eFP browser' database and provided more description for 'The plant public RNA-seq database'. Besides, we have also provided the detailed information of the databases used for LAO1 expression pattern analysis (Figure EV1C-E) in Appendix Table S1 of the revised manuscript.

Suppl fig 4. There are two #1 on figure 4A. can the authors explain?

Author Reply: Thank you for pointing this out. The newly developed leaf is too small to provide enough tissue for total protein extraction. Therefore, we define the newly developed leaf along with the adjacent leaf as Group #1 in our experiments. We have added more detailed descriptions about how the groups were defined in the legends of the revised Figure EV2A.

Supplemental figure 6, the seedlings are rather small in the photos, can the authors increase the magnification and the resolution of the images shown?

Author Reply: The relevant data from Supplemental Figure 6 have been moved to Figure 3F in the revised manuscript, but only the data of -N 24h were shown and quantified. We have repeated the experiments and, as suggested, we have included images with higher magnification and resolution in the revised Figure 3F.

Referee #2:

Li et al., identified a novel module that involves an F-box protein LAO1, a ubiquitin receptor protein DSK2, and class I TCP transcription factors regulating Arabidopsis response to nitrogen deficiency and drought tolerance. While the paper is interesting, well-written, and generally robust in its experimental approach, some additional experimental validation or clarification of the conclusions is necessary to strengthen it further and to clarify the link between the regulation of drought and nitrogen stress

responses by the identified regulatory module

Author Reply: We thank the reviewer for the positive evaluation and critical comments on our findings.

1) I suggest that the authors include a schematic model for visualizing their findings.

Author Reply: This is a good suggestion. As shown in Figure EV5, we have included a schematic model to make it easier for readers to understand the findings.

2) I am missing any direct evidence that LAO1 can be recruited to autophagosomes via DSK2 and degraded by autophagy. The authors could assess this using several relatively straightforward assays, especially since many of the necessary transgenic lines are already available in their labs.

- Assess co-localization of mCherry-LAO1 and GFP-ATG8 in both Col-0 and *dsk2 cr* mutant background under control/ nitrogen starvation conditions (using ConA treatment)
- Assess co-localization of mCherry-LAO1 and GFP-DSK2 under control/ nitrogen starvation conditions (using ConA treatment)
- Detect the transport of the tagged LAO1 to the vacuole under control/nitrogen starvation (using ConA treatment)

Author Reply: Thank you very much for your valuable comments and suggestions. We have conducted the recommended experiments to address your concerns.

Firstly, we assessed the co-localization of mCherry-LAO1 and GFP-ATG8 in protoplasts isolated from both wild-type (Col) and *dsk2 cr* mutant background under both +N and -N (ConA) conditions as shown below. We did observe that after -N with ConA treatments, mCherry-LAO1 and GFP-ATG8A could colocalize in puncta in Col protoplasts but not *dsk2 cr* protoplasts. Despite these observations, we encountered significant technical challenges in finding protoplasts with both mCherry and GFP signals, which made it difficult to quantitatively analyze the data. Therefore, we have not included this result in the revised manuscript at this stage.

Secondly, we assessed the colocalization of mCherry-LAO1 and GFP-DSK2 in the revised Figure 4E. After -N with ConA treatments, mCherry-LAO1 could colocalize with GFP-DSK2 in puncta within the cells.

Thirdly, we investigated the localization of GFP-LAO1 using 35S: GFP-LAO1 transgenic seedlings. As shown in Figure 3E, the GFP signals accumulated in the vacuole under nitrogen starvation conditions in the presence of Con A.

Collectively, these results provide direct evidence that LAO1 is recruited by DSK2 into autophagosomes for degradation.

3) Mutations in DSK2 lead to better performances in both Col and HA-LAO1 backgrounds after nitrogen starvation (Figure 5E). This is in part due to the link between DSK2 and TCPs. However, could the author rule out the possibility of the reduced expression of LAO1 (and therefore its protein level) in the *dsk2 cr* mutants? (e.g., in case DSK2 controls the transcriptional regulator of LAO1 as an alternative regulatory mechanism). For a comprehensive assessment, the authors should measure the transcript abundance of

LAO1 in *dsk2 cr* mutants (under control and nitrogen deficiency conditions).

Author Reply: Thanks for raising the hypothesis. As suggested, we have conducted an analysis of *LAO1* transcript abundance in both Col and *dsk2 cr* mutants under both +N and -N conditions. As shown in Fig. EV3D, mutations in *DSK2* did not result in a reduction of *LAO1* mRNA levels. This finding excludes the possibility that the improved performance of *dsk2* mutant is attributed to the lower expression level of *LAO1*.

4) How does DSK2 mediate the degradation of TCPs? Is there evidence to include or rule out the involvement of autophagy in this process?

Author Reply: Thanks for your question. To this end, we treated the protoplasts transfected with GFP-DSK2/TCPs-LUC with MG132 or E64d, as shown in the revised Figure 6F. The results showed that both MG132 and E64d treatments could increase the luminescence ratio of TCPs-LUC/REN, indicating that they can both inhibit DSK2-mediated degradation of TCPs. These results indicate that both ubiquitin-proteasome and autophagy pathways participate in DSK2-mediated TCP degradation.

5) Does the DSK2-mediated degradation of TCPs occur during nitrogen starvation as well?

Author Reply: Thanks for your question. We performed western blots to monitor GFP-TCP14 and GFP-TCP21 protein levels in both Col and *dsk2 cr* mutant backgrounds following nitrogen starvation. As shown in Figure 6G, the protein levels of GFP-TCP14 and GFP-TCP21 were consistently higher in the *dsk2 cr* mutant backgrounds under normal nitrogen conditions. Moreover, nitrogen starvation induced a significant increase in the protein levels of both GFP-TCP14 and GFP-TCP21, with this increase being more pronounced in the *dsk2 cr* mutant background. These observations suggest that DSK2-mediated degradation of TCPs occurs during nitrogen starvation as well.

6) Is the effect of TCPs mitigating LAO1 specific to N starvation, or does it also apply to drought? To address this, the authors could test the phenotypes of HA-LAO1/GFP-TCP14/ and HA-LAO1/GFP-TCP21 in response to drought conditions.

[REDACTED: Authors' response with unpublished data.]

Figure for referees not shown.

7) Please show the phenotype of 35S: HA-LAO/atg5 (and atg7-3) mutant analyzed in Figure 3C, especially after 24h of N deficiency. I assume these plants barely survive the stress.

Author Reply: We previously presented these phenotypes in Supplemental Figure 6 in the last version of the manuscript. In the revised manuscript, as suggested, we have shown and quantified the phenotypes of 35S: HA-LAO1/atg5 and 35S: HA-LAO/atg7 after 24 hours of N starvation treatments in Figure 3F. As expected by the reviewer, these two lines exhibited significant growth defects as compared to atg5, atg7, and 35S:HA-LAO1. These findings are consistent with the immunoblot analysis presented in Figure 3B and 3C, further supporting the role of LAO1 in nitrogen starvation responses and the role of autophagy pathway in LAO1 protein degradation.

Minor comments:

1. Figure 3A: the immunoblotting data should be quantified, especially after 60 min.

Author Reply: We have repeated the experiments and quantified the results in the revised manuscript, as suggested by the reviewer.

2. Line 167 describes the experimental setup as "followed by a two-day recovery on 1/2 MS plates," while in the legends of Figures 1D& E, a four-day recovery is mentioned.

Please clarify this inconsistency.

Author Reply: Thanks for pointing out this inconsistency. Different genotypes exhibit varying extents of growth phenotypes when recovering after nitrogen starvation, so we used different recovery periods for different genotypes. For this particular case, we used a two-day recovery period for the HA-LA01 experiments, while a four-day recovery period for the *lao1* mutant experiments. To avoid any further misunderstanding, we have specified the treatment condition for each experiment in the legends of the respective figures in the revised manuscript.

3. Figure 7B: It is unclear which -N condition presented in panel A is quantified in B? Is it -N 6d, R5d or -N 7d, R4d?

Author Reply: Thanks for pointing this out. In Figure 7B, we quantified the data from the -N 7d, R4d condition shown in Figure 7A. We apologize for any confusion this may have caused and have included this information in the legends of the revised manuscript.

Col-0: I suggest using the standard format, 'Col-0,' rather than just 'Col.' Or specify 'Col-0' the first time it is mentioned in the main text to indicate which Columbia accession is being referred to.

Author Reply: As suggested, we have specified 'Col-0' the first time it is mentioned in the manuscript. Lines 143-144.

Referee #3:

The manuscript presents a considerable amount of working looking at the potential role of

the F-box protein LAO1, the pexophagy receptor DSK2 that binds to and target proteins for autophagic degradation, and transcription factors from the TCP family, in responses to nitrogen limitation and drought stress.

Author Reply: We thank the reviewer for the evaluation of this study. Your feedback is greatly appreciated.

The title of the manuscript is not supported by the results presented. The connection between the proteins of interest, nutrient stress and drought stress is not made. Analysis of involvement in drought stress response is very limited, with no results shown for LAO1 involvement in drought tolerance.

Author Reply: Thank you for the valuable suggestions. We have revised the title of the manuscript to: 'DSK2 modulates degradation of F-box protein LAO1 and class I TCPs for nitrogen starvation response'. In addition, we have removed the drought aspect from the revised manuscript.

What is meant by -N? The material and method section does not include information on N concentrations in growth media. This is a significant omission. Does the media completely lack N or is N limited? If limited, is it mild or severe N limitation? As the authors may be aware, plants respond differently depending on the severity of N limitation (see Plant physiology 166(2) 509-517; Molecular Plant, 15(1), 86-103). What is the source of N?

Author Reply: Thank you for pointing this out. The -N condition in our study indicates a complete lack of macronutrient nitrogen in the medium. The catalog number of the MS basal salts lacking nitrogen was listed in the Reagents_Tools_Table. For the control, we used normal MS medium containing both nitrate and ammonium as the N sources. We have added this information in the Methods part of the revised manuscript. Lines 536-538

Figure 2C - While the level of LAO1 is lower under -N compared to +N at each timepoint,

the level of LAO1 seems to be increasing over time under both +N and -N conditions.

Author Reply: We agree with the reviewer that LAO1 protein levels increase gradually as the young seedlings grow in Figure 2C. Actually, we consistently observed this phenomenon, as shown in Figure EV 2D and 2F. Furthermore, we provided evidence that LAO1 protein levels are positively associated with nitrogen assimilation process in vivo (Figure EV 2G-J). This association may contribute to the increase in LAO1 protein levels as young seedlings grow and develop true leaves, a stage when N assimilation is particularly active.

The degradation rate of LAO1 under -N needs to be quantified and compared to that under +N (Figure 3A-C). To evaluate the role of DSK2 in modulating LAO1 abundance, CHX assays should be used to examine the degradation of LAO1 in the presence and absence of DSK2 under +N and -N conditions.

Author Reply: Thank you for the comments. As shown in Appendix Figure S2, the degradation rate of LAO1 in the absence of nitrogen is significantly faster than that in the presence of nitrogen. The results indicate that LAO1 protein is predominantly degraded following nitrogen starvation, after which autophagy pathway is activated. Therefore, we believe that it is more appropriate to focus on nitrogen-starvation-induced LAO1 degradation when examining the effects of E64d, *atg* mutation, or *dsk2* mutations.

As suggested, we have repeated and quantified all the experiments related to the degradation rate analysis of LAO1 in the absence of nitrogen (Figures 3A-C, 4F, and 4G). We also modulated the degradation rate of LAO1 in both Col and *dsk2* cr mutant backgrounds under -N condition in the presence of CHX (Figure 4F). These results support the conclusion that DSK2 plays a crucial role in modulating LAO1 protein abundance.

Figure 5E, loss of DSK2 should lead to the overaccumulation/high levels of LAO1. - should the *dsk2* mutant not look similar to the LAO1 overexpresso (OE) ? And should the *dsk2* LAO1 OE be more sensitive to -N conditions compare to Col-0 and LAO1 OE? Quantification is needed for the seedlings in Figure 5E. Similar experiments are required for examining the impact of DSK2 on TCP stability.

Author Reply: We agree with the reviewer that mutations in *DSK2* should have resulted in a *LAO1*-OE phenotype. However, we observed the opposite in Figure 5E. As suggested, we have quantified the phenotypes in the revised manuscript (Figure 5F). The *dsk2* cr mutants consistently exhibited slightly better performance as compared to the wild type, which is contrary to the phenotypes of *LAO1*-OE. The unexpected phenotypes have led us to further elucidate the regulatory network of *DSK2*, identifying class I TCPs as the targets of *DSK2* in vivo. Class I TCPs play a positive role in plant tolerance to nitrogen starvation, and their proteins also accumulate in the *dsk2* cr mutant background. More importantly, our results support that class I TCPs can mitigate the effect of *LAO1*-OE in response to nitrogen starvation (Figure 7C-F). We believe that the accumulation of TCP proteins in the *dsk2* cr mutant contributes to the nitrogen-starvation tolerant phenotypes of the *dsk2* cr mutants.

As suggested, we also performed similar experiments to examine the impact of *DSK2* on TCP stability in the Appendix Figure S5. Both the *dsk2* mutation and *TCP* overexpression in Col background improved plant performance after nitrogen starvation treatment, supporting the roles of *DSK2* and *TCP* in plant adaptation to nitrogen starvation. However, the combination of *dsk2* mutation and *TCP* overexpression did not additively enhance the phenotype. Given that *DSK2* can control the degradation of both *LAO1* and *TCP*, two factors with opposite function, we propose that the accumulation of *LAO1* protein following *dsk2* mutation will also interfere with *TCP* function. Consequently, *DSK2*, *LAO1*, and *TCP* form a network that balances plant adaptation to nitrogen starvation and growth.

Does TCP regulate the expression of *LAO1* and/or *DSK2*?

Author Reply: Thank you for this question. We performed RT-qPCR analysis of *DSK2* and *LAO1* in multiple *tcp* mutants (Figure EV4A). The results demonstrate that *TCP* does not regulate the transcription of *LAO1* or *DSK2*.

Dear Dr. Dong

Thank you for the submission of your revised manuscript to EMBO reports. We have now received the full set of referee reports that is copied below.

As you will see, all referees state that the study has been significantly strengthened during the revision. That said, the referees also raise a number of remaining concerns, that will need to be resolved. Please address the remaining concerns from all three referees and please provide a point-by-point response. Your revised manuscript will be sent back to referee #3 for a final evaluation.

From the editorial side, there are also a few things that I kindly ask you to address:

- Please reduce the number of keywords to 5.
- References: et al is used after 10 author names instead of 8. Please update the format accordingly.
- Author Checklist: please complete column D, rows 81-88, by choosing the appropriate answer from the pull-down menus.
- Please add callouts for Figure 5F, Appendix Table S3, and Appendix Table S4 in the text.
- Appendix: please add page numbers to the table of content and upload the file as PDF. May I in addition suggest moving the first figure to the next page, to have the table of content on a dedicated "title page"?
- Appendix Figures S6, S7 and S8 lack a descriptive legend.
- If publicly available data were reused, data citations should be used. To align to this format, you cite the paper that reported on the generation of the datasets, e.g., Alvarez JM et al YEAR and in addition the dataset itself. For PRJNA555731 this would look like this:
In the text: (Alvarez JM et al, YEAR, Data ref: Alvarez JM et al, YEAR).
In the reference list:
Alvarez JM..., YEAR
Alvarez JM... YEAR, URL linking to the dataset [DATASET]
- I understand that you have reanalysed a large number of datasets. Maybe you could refer to the original publications in the methods section and then change Appendix Table S1 into a Dataset .xls file that lists the Project number, the Group Information, the URLs to the Datasets and the reference?
- Accession Numbers: you state that "Sequence data from this article can be found in the TAIR...". The phrase leaves it unclear whether these sequencing data were de novo generated by you or whether you referred to/used this data. Can you please clarify?
- Reagents and Tools Table: Please remove the Instructions paragraph from the table.
- We perform a routine figure and data integrity check on all revised manuscripts. Doing so we noticed that the images shown in Figure 7E, for "tcpQ" and "lao1 cr/tcpQ" in the -N condition, seem to be the same. In the source data you supplied, the specimens look different, so I assume this was a copy-paste mistake. Please check.
- The Western blots shown in Appendix Fig S2 and Appendix Fig S6 seem to be the same. The legend in Appendix Fig S6 states that the protein level is related to Appendix Figure S2, but does not explicitly call out the reuse. Is the reuse necessary? If so it needs to be called out in the legend. "Blot ___ reused in appendix Figure S6."
- As outlined above, panels S6, S7, and S8 need a descriptive legend.
- The YFP panels shown in Figure EV4C appear to lack any signal. To avoid potential unambiguities, please provide the source data for this panel to confirm the image signal. Thank you.
- Please upload the source data as one folder per figure. Inside each folder, the files should be organized in subfolders, one subfolder for each panel. It would be better to supply the images as .tiff, .png, .jpg files instead of incorporating them in the .xls files.
- Our production/data editors have asked you to clarify several points in the figure legends (see below). Please incorporate these changes in the manuscript and return the revised file with tracked changes with your final manuscript submission.

A) Statistical test information. Only p-values that are actually shown in the figure panel(s) should (and must) be defined in the legends, all others should be removed from (or added to) the legend. Moreover, we ask for the specification of exact p-values:
- Please indicate what */ **/ ***/ ****/a /a' /b represents; if this represents p value(s), please specify the exact p value in the legend(s) of figure(s) 1E, G; 3A, B, C, F, H; 4F, G; 5B, F; 6F, G; 7B, D, F; EV2 B; 3D.
- Please note that the exact p values are not provided in the legends of figures 2A, D, E, 3D, 4H; 6D, E."

- As a standard procedure, we edit the title and abstract of manuscripts to make them more accessible to a general readership. Please find the edited versions below my signature.

- Finally, EMBO Reports papers are accompanied online by
A) a short (1-2 sentences) summary of the findings and their significance,
B) 2-3 bullet points highlighting key results.
Please send us this information along with the revised manuscript.

I look forward to seeing a revised form of your manuscript when it is ready.

Yours sincerely,

=====

Referee #1:

This second version of the manuscript has considerably improved, and most of my doubts were solved. However I think that the title could be slightly amended o better reflect the manuscript, maybe in "DSK2 modulates degradation of F-box protein LAO1 and class I TCPs in nitrogen starvation response".

In addition the model is not very clear and I believe an extra care should be taken while drawing it. It should reflect more the conclusions of the manuscript, and the interconnected molecular functions of LAO1, DSK2, and TCPs.

The resolution of some of the photos, such as the photo presented in Figure 1B, is very low and should be improved.

Referee #2:

The revised manuscript is significantly improved, and all my comments have been well addressed. I have only one additional comment to the authors: please include information about how early (or late) after transfer to -N medium and ConA the confocal microscopy shown in Figure 3E was conducted. I am asking because in the image, I do not see LAO1-GFP as puncta within the vacuole; instead, the signal appears more diffuse, which suggests it may represent a later stage of the process (?).

Referee #3:

The authors have addressed many of the reviewers comments. The revised manuscript includes substantial changes that strengthen the conclusions, linking DSK2, LOA1 and TCP to plant response to nitrogen starvation. However, with the revised manuscript brings a number of issues that should be addressed.

The authors state that "Immunoblot analyses showed that 48 and 60 hours of nitrogen starvation significantly reduced HA-LAO1 protein levels" (line 109). LOA1 levels is lower under -N compared to +N (Figure 2C and D). However, the blot shown suggests increase in protein abundance over time under both +N and -N conditions, compared to the 24 hour time point (this issue was not adequately addressed in the response). As expression is driven by the 35S promoter, this suggest that both -N and +N stabilize LAO1, hence the increase in protein abundance. This does not correlate with the findings from the CHX chase assays to monitor protein stability (Appendix Fig. S2), which shows degradation (instability) of LAO1 under both +N and -N. It is noted that degradation of LOA1 seems more pronounced under -N, however this needs to be quantified over multiple blots. The regulation of LAO1 abundance does not seem to be as simple as indicated, where -N inhibit expression/reduce abundance. Also, what is the difference between blots shown in Figure S2 and S6? S6 is not mentioned in the manuscript.

The GFP signal is difficult to see in Figure 2F

Figure 3E require a -N control image.

Figure 3B and 3C - the degradation rate of HA-LAO1 in each background (with and without ATG5/7) should be compared to

determine the impact of loss of ATG5/7 on LAO1 turnover. In the graphs shown the relative level of HA-LAO1 in the atg mutant backgrounds are compared to the 0 timepoint of wildtype background. To demonstrate differences in degradation over time, each timepoint in a series should be compared to its own 0 (i.e. the 0 timepoint within that series). This also applies to the graphs shown in figure 4F and 4G.

Which DSK2 isoform was used for the interaction assays? For Figure EV3C, is the antibody expected to detect both isoforms of DSK2 as shown in *The Plant Cell* 22.1 (2010): 124-142. If not, which isoform is detected?

DSK2 crisper - Figure S3 indicates a 242bp deletion for DSK2A #13, is this correct?

Line 384 states "Overexpression of TCP14 and TCP21 improved plant fitness to nitrogen starvation (Appendix Fig. S5)." The TCP overexpressors are not shown in figure S5 and I could not find this data in any other figure. Assessment of the response of TCP overexpressor to -N is needed, especially since a number of double overexpressors using TCP (e.g. GFP-TCP21 HA-LAO1 (Fig. 7C and D)) is discussed in the manuscript.

The authors suggest "that DSK2 may also control the protein level of a factor that antagonizes the effect of LAO1 protein accumulation" (line 344). This is based on the finding that the DSK2 loss of function mutant is unexpectedly, and interestingly, more tolerant of nitrogen starvation conditions (Figure 5E). This led to the identification of TCPs as DSK2 interactors, and the increase abundance of the TCPs in dsk2 mutant may explain the increase tolerance to N starvation shown in figure 5E. How does the dsk2 tcp double mutant respond to nitrogen starvation compared to the dsk2 single mutant? If the increase in TCP levels in dsk2 mutant is responsible for the increase -N tolerance, then loss of TCP in the dsk2 background should reduce tolerance. Also, the link between DSK2, TCP and -N is based only on the increase in TCP abundance in dsk2 mutants under -N conditions. In addition to assessing the response of dsk2 tcp double mutant seedlings/plants to -N, the authors may consider looking at overexpressing TCP in the dsk2 background (similar to experiment done for LAO1 [Figure 5]).

=====

DSK2-mediated degradation of F-box protein LAO1 and class I TCPs modulates the nitrogen starvation response

Plants have evolved intricate strategies to cope with various abiotic stresses. Ubiquitin-mediated protein degradation plays a key role in plant development as well as abiotic stress tolerance. In this study, we identify LAO1, an F-box protein with unknown function, as a negative regulator of plant fitness during nitrogen starvation. DOMINANT SUPPRESSOR OF KAR 2 (DSK2) interacts with and mediates the autophagic degradation of LAO1 protein during nitrogen starvation. The loss of LAO1 improves the fitness of an autophagy-deficient mutant, atg5-1, under nitrogen starvation. Intriguingly, mutations in DSK2 facilitate rather than reduce plant growth after nitrogen starvation. This unexpected effect of DSK2 knockout led us to discover that DSK2 also interacts with and degrades a group of class I TCPs transcription factors. Phenotypic observations demonstrate that class I TCPs are crucial for plant adaptation to nitrogen starvation. Moreover, genetic analyses indicate that class I TCPs function downstream of LAO1 and counteract its negative effects. Collectively, our findings unveil a previously undescribed regulatory network governing plant fitness during nitrogen starvation.

Referee #1:

This second version of the manuscript has considerably improved, and most of my doubts were solved. However I think that the title could be slightly amended o better reflect the manuscript, maybe in "DSK2 modulates degradation of F-box protein LAO1 and class I TCPs in nitrogen starvation response".

Author reply: Thank you for the positive evaluation of our revised manuscript and for your valuable suggestions. We have carefully considered your feedback and have amended the title to “DSK2-mediated degradation of F-box protein LAO1 and class I TCPs modulates the nitrogen starvation response”. This revised title better reflects the content of the manuscript and is more accessible to a general readership. We appreciate your guidance in improving the clarity and impact of our work.

In addition the model is not very clear and I believe an extra care should be taken while drawing it. It should reflect more the conclusions of the manuscript, and the interconnected molecular functions of LAO1, DSK2, and TCPs.

Author reply: Thank you for your valuable feedback regarding the clarity of the model. We have taken extra care to improve the model, ensuring that it better reflects the conclusions of the manuscript and the interconnected molecular functions of LAO1, DSK2, and TCPs in the context of nitrogen starvation-influenced plant growth. We believe these revisions will enhance the clarity and comprehensiveness of the model, making it more informative for our readers. We appreciate your suggestions and are confident that these changes will improve the overall quality of our manuscript.

The resolution of some of the photos, such as the photo presented in Figure 1B, is very low and should be improved.

Author reply: Thank you for pointing out this issue. To facilitate review process, we reduced the file size, which may have affected the quality of the images. We have thoroughly checked the original figures and can make sure that we have all the high-resolution photos available. We will ensure that the final version of the manuscript includes these high-resolution images to improve clarity and quality. We appreciate your attention to this detail.

Referee #2:

The revised manuscript is significantly improved, and all my comments have been well addressed. I have only one additional comment to the authors: please include information about how early (or late) after transfer to -N medium and ConA the confocal microscopy shown in Figure 3E was conducted. I am asking because in the image, I do not see LAO1-GFP as puncta within the vacuole; instead, the signal appears more diffuse, which suggests it may represent a later stage of the process (?).

Author reply: Thank you for the positive evaluation of our revised manuscript and for your additional comments. We have added detailed information about the timing of the treatments in the legends of Figure 3E. Specifically, seven-day-old seedlings were pretreated with normal 1/2 MS (+N) or nitrogen-free nitrogen-free 1/2 MS (-N) medium for 48 h,

followed by treatment with 1 μ M ConA for 3 h.

We agree with you that the signal appears more diffuse, albeit there are puncta-form of LAO1-GFP signals in these images. The appearance of the signal can be influenced by several factors, including the protein itself, the tissue type, and the stage of the process. We appreciate your attention to this detail and hope that the additional information clarifies the timing and interpretation of our results.

Referee #3:

The authors have addressed many of the reviewers comments. The revised manuscript includes substantial changes that strengthen the conclusions, linking DSK2, LOA1 and TCP to plant response to nitrogen starvation. However, with the revised manuscript brings a number of issues that should be addressed.

Author reply: Thank you for the evaluation of the revised manuscript and your valuable comments. We are pleased that our revisions have strengthened the conclusions linking DSK2, LAO1, and TCPs to plant response to nitrogen starvation. We appreciate your detailed feedback and are committed to addressing the issues you have identified.

The authors state that "Immunoblot analyses showed that 48 and 60 hours of nitrogen starvation significantly reduced HA-LAO1 protein levels" (line 109). LOA1 levels is lower under -N compared to +N (Figure 2C and D). However, the blot shown suggests increase in protein abundance over time under both +N and -N conditions, compared to the 24 hour time point (this issue was not adequately addressed in the response). As expression is driven by the 35S promoter, this suggest that both -N and +N stabilize LAO1, hence the increase in protein abundance. This does not correlate with the findings from the CHX chase assays to monitor protein stability (Appendix Fig. S2), which shows degradation (instability) of LAO1 under both +N and -N. It is noted that degradation of LOA1 seems more pronounced under -N, however this needs to be quantified over multiple blots. The regulation of LAO1 abundance does not seem to be as simple as indicated, where -N inhibit expression/reduce abundance. Also, what is the difference between blots shown in Figure S2 and S6? S6 is not mentioned in the manuscript.

Author reply: We sincerely appreciate your careful examination of our data and the insightful comments regarding the regulation of LAO1 protein abundance under nitrogen starvation conditions. We agree that the regulation of LAO1 is complex and have addressed the concerns raised by the reviewer as follows:

1) We acknowledge that the blot shown in Figure 2C suggests a significant increase of HA-LAO1 protein over time under +N condition, with a very mild increase under -N condition. We would like to clarify that the result does not conflict with the CHX assays in Appendix Fig. S2, which show that N starvation reduces HA-LAO1 protein stability. Instead, it suggests a complex regulation of LAO1 protein abundance. In Figure EV2F, we observed that HA-LAO1 protein levels are also regulated by developmental stage, correlating with N assimilation and true leaf development. This result suggests that sufficient N supply maintains the accumulation of HA-LAO1 during development, while this accumulation is absent under -N condition as shown in Figure 2C. We believe that this result strongly supports the idea that -N interferes with HA-LAO1 protein accumulation during

development, implying a negative effect of -N on LAO1 protein abundance. We agree with you that the regulation of LAO1 abundance is complex, and have revised the text to more accurately describe the results shown in Figure 2C (Lines 199-206) as well as to provide a more detailed discussion of these findings (Lines 461-470).

2) As you suggested, we repeated the CHX treatment assays shown in Appendix Fig. S2, and quantified the results across multiple independent experiments. The updated results consistently confirm that LAO1 degradation is significantly faster under -N conditions, supporting our conclusion.

3) In the previous version of the manuscript, Appendix Figure S6 is the source data of Appendix Figure S2. In the revised manuscript, we have provided all the source data as individual files, and have updated the Appendix files to avoid confusion. We apologize for any inconvenience this may have caused.

The GFP signal is difficult to see in Figure 2F

Author reply: Thank you for pointing this out. We have improved Figure 2F to clearly show the GFP signal.

Figure 3E require a -N control image.

Author reply: Thank you for pointing this out. We have added -N control image in Figure 3E. The -N control consistently show a reduction in GFP-LAO1 signal, as observed in Figure 2F.

Figure 3B and 3C - the degradation rate of HA-LAO1 in each background (with and without ATG5/7) should be compared to determine the impact of loss of ATG5/7 on LAO1 turnover. In the graphs shown the relative level of HA-LAO1 in the atg mutant backgrounds are compared to the 0 timepoint of wildtype background. To demonstrate differences in degradation over time, each timepoint in a series should be compared to its own 0 (i.e. the 0 timepoint within that series). This also applies to the graphs shown in figure 4F and 4G.

Author reply: Thank you for your valuable comments. As you suggested, we have re-plotted the relative degradation rate of HA-LAO1 in each background using their own start timepoint as control for Figure 3B and 3C, as well as Figure 4F and 4G. The revised plots are shown in Appendix Figure S3. The conclusions drawn from these revised plots are consistent with those from the main figures.

We understand the importance of comparing each timepoint to its own 0 timepoint to accurately demonstrate differences in degradation over time. The previous graphs in the main figures were intended to show both the differences in degradation rate and the HA-LAO1 protein levels in *atg5/atg7/dsk2* mutant background as compared to the wild-type control Col at each time point. While we kept using the bar graphs in the main figures, we have included the revised plots in Appendix Figure S3.

Which DSK2 isoform was used for the interaction assays? For Figure EV3C, is the antibody expected to detect both isoforms of DSK2 as shown in The Plant Cell 22.1 (2010): 124-142. If not, which isoform is detected?

Author reply: Thank you for your question. We used DSK2B for the interaction assays. To

better convey this information, we have thoroughly reviewed the figures and specified which DSK2 isoform was used in all relevant figures (Figures 4, 5, EV3, and EV4).

Regarding to the DSK2 antibody, we used full-length DSK2B protein for antibody generation. Based on the 87% amino acid sequence identity of DSK2A and DSK2B, we highly suspected that this antibody could detect both isoforms. To verify this, we attempted to isolate *dsk2a* or *dsk2b* single mutants but were only successful in obtaining a *dsk2a cr* mutant. As shown below, the DSK2 antibody detected a lower level of DSK2 signal in *dsk2a cr* mutant and a complete loss of DSK2 signal in *dsk2 cr#1* mutant as compared to Col. Overall, these results suggest that the DSK2 antibody can detect both DSK2A and DSK2B, and the signals accurately reflect the endogenous protein levels of DSK2 *in vivo*.

DSK2 crisper - Figure S3 indicates a 242bp deletion for DSK2A #13, is this correct?

Author reply: Thank you for pointing this out. Yes, the previous Appendix Figure S3, which is now the revised Appendix Figure S4, correctly indicates a 242 bp deletion for DSK2A in *dsk2cr* #13. We have updated the figure to better reflect the mutation types and have included the alignment results below and in source data to support this.

```

CLUSTAL format alignment by MAFFT (v7.511)

DSK2ref      tagccocatcagctatttctcatcatttcacgcataagctctgggtttctgcgcgcttcta
DSK2-13A     tagccocatcagctatttctcatcatttcacgcataagctctgggtttctgcgcgcttcta
*****
sgRNA
DSK2ref      gagtttgacgaaggatgcttgggtcattgagcacatgaccaagctcaggattgogactca
DSK2-13A     gagtttgacgaaggat-----
*****

DSK2ref      caagctcagcatttgagggttattcataatcatactcctcataaattotgggtgttca
DSK2-13A     -----

DSK2ref      taagattctggatggctggagtattcatcatctctaatcatggttagggtctgagcta
DSK2-13A     -----

DSK2ref      gttgtgtctgtgctgtcagatcaggaagaccagatccaaataatccagccatagcat
DSK2-13A     -----

DSK2ref      ttccaccacccaaaggattaaatccaaggccaggaacaaagattcacctcctcogaggt
DSK2-13A     -----taaatccaaggccaggaacaaagattcacctcctcogaggt
*****

DSK2ref      ttgagctatcatttgaaccaaccogttgaggagcagtggttggttccagcatttgca
DSK2-13A     ttgagctatcatttgaaccaaccogttgaggagcagtggttggttccagcatttgca
*****

DSK2ref      cgggagcagaaggagaagaaggcacaaaacctogaacatataacgggtgtgatcagcct
DSK2-13A     cgggagcagaaggagaagaaggcacaaaacctogaacatataacgggtgtgatcagcct
*****

DSK2ref      gcaaacctatatataaaatggtccacagagaaggaaacattgaggctcatga
DSK2-13A     gcaaacctatatataaaatggtccacagagaaggaaacattgaggctcatga
*****

```

Line 384 states "Overexpression of TCP14 and TCP21 improved plant fitness to nitrogen starvation (Appendix Fig. S5)." The TCP overexpressors are not shown in figure S5 and I could not find this data in any other figure. Assessment of the response of TCP overexpressor to -N is needed, especially since a number of double overexpressors using TCP (e.g. GFP-TCP21 HA-LAO1 (Fig. 7C and D)) is discussed in the manuscript.

Author reply: Thank you for your valuable comments. We apologize for any oversight in the previous version of the manuscript. We have now included the data for the TCP14 and TCP21 overexpressors in the revised Appendix Fig. S6 (previously Appendix Fig. S5).

In addition to the phenotype data shown in the previous Appendix Fig. S5, we have further added RT-qPCR results for *TCP14* and *TCP21* in the revised Appendix Fig. S6. The overexpression lines TCP14 OE#32 and TCP21 OE#77 are included, and their responses to nitrogen starvation are clearly demonstrated. Thank you for your consideration of these revisions.

The authors suggest "that DSK2 may also control the protein level of a factor that antagonizes the effect of LAO1 protein accumulation" (line 344). This is based on the finding that the DSK2 loss of function mutant is unexpectedly, and interestingly, more tolerant of nitrogen starvation conditions (Figure 5E). This led to the identification of TCPs as DSK2 interactors, and the increase abundance of the TCPs in *dsk2* mutant may explain the increase tolerance to N starvation shown in figure 5E. How does the *dsk2 tcp* double mutant respond to nitrogen starvation compared to the *dsk2* single mutant? If the increase in TCP levels in *dsk2* mutant is responsible for the increase -N tolerance, then loss of TCP in the *dsk2* background should reduce tolerance. Also, the link between DSK2, TCP and -N is based only on the increase in TCP abundance in *dsk2* mutants under -N conditions. In addition to assessing the response of *dsk2 tcp* double mutant seedlings/plants to -N, the authors may consider looking at overexpressing TCP in the *dsk2* background (similar to experiment done for LAO1 [Figure 5]).

Author reply: We sincerely appreciate the reviewer's insightful suggestions, which have significantly strengthened our study. In response to the comments, we have conducted additional experiments to further elucidate the relationship between DSK2, TCPs, and nitrogen (N) starvation response.

Following your suggestion, we examined the phenotypes of Col, *dsk2 cr*, *tcpQ*, and *dsk2 cr tcpQ* under N starvation conditions. As shown in the Appendix Figure S7, the *dsk2 cr tcpQ* mutants exhibited a similar phenotype to the *tcpQ* mutant, supporting our hypothesis that TCP accumulation is indeed responsible for the improved N starvation tolerance observed in the *dsk2 cr* mutant. The results provide compelling genetic evidence for the functional link between DSK2, TCPs, and N starvation.

In response to your previous request "Similar experiments are required for examining the impact of DSK2 on TCP stability." in the context of N starvation phenotypes, we have previously provided the phenotypic data for Col, *dsk2 cr*, *TCP14* OE, *TCP14* OE/*dsk2cr*, *TCP21* OE, and *TCP21* OE/*dsk2cr* in Appendix Fig. S5 (now updated to Appendix Fig. S6 in this revised manuscript). Consistent with the conclusions of our earlier findings, both the *dsk2* mutation and TCP overexpression in Col background improved plant performance under nitrogen starvation treatment. In line with TCPs and DSK2 function in the same genetic pathway regulating N starvation response, TCP

overexpression did not additively enhance the *dsk2 cr* mutant phenotypes. This phenomenon can be explained by two possible mechanisms: 1) Mutations in *DSK2* likely lead to the accumulation of multiple TCP proteins (Figure 6E), which may collectively contribute to the observed phenotype. In this context, overexpression of a single *TCP* gene might be insufficient to significantly enhance the effect beyond what is already achieved by the coordinated action of multiple TCPs accumulating in the *dsk2 cr* mutant. 2) The N starvation response pathway regulated by DSK2-TCP may have reached its maximum capacity in the *dsk2 cr* mutant background. Therefore, additional overexpression of a single *TCP* gene cannot further amplify the physiological response.

These additional experiments have significantly strengthened our conclusions and provided deeper insights into the molecular mechanisms underlying N starvation tolerance. We believe these findings have substantially improved the manuscript and addressed your concerns.

Dear Dr. Dong

Thank you for the submission of your revised manuscript to EMBO reports. We have now received the report from the referee who was asked to assess it.

As you will see, referee 3 acknowledges that the study has been significantly strengthened during the second round of revision but s/he also raises a few remaining points that I kindly ask you to address.

There are also three things I would still need from the editorial side:

1) Data references:

For each dataset you analyzed we would need the following:

In the text you would have to cite both, the primary paper and the dataset. Thus, each of these references for Fig EV1 would have to look like this:

(Alvarez et al. 2020, Data ref: Alvarez et al., 2020; Soto-Cardinault et al., 2023, Data ref: Soto-Cardinault et al., 2023) etc.

The same in the reference list:

You would need first the citation to the paper of e.g. Alvarez et al

And then in addition the citation of the dataset, e.g., Alvarez JM,... YEAR [URL linking to the dataset] and this individually for every dataset used.

I feel that this is getting too complicated in your case.

I therefore suggest citing only the original paper in the manuscript itself, but refer to Dataset EV1 for the datasets analyzed.

Would this make sense?

2) Could you please provide the raw, individual figure files for Fig EV4C in addition to the compiled figure? Thank you very much!

3) In the source data for Figure 1G, the chlorophyll content measured for HA-LAO1 #1 and #2 under -N conditions show the exact same numbers (Column E, line 6 and 7; Column F, line 6 and 7). Can you please check whether this is correct? The same is true for Figure 3H, column D, line 4 and 5). These numbers are the same up to the 6th decimal, which appears unusual.

With kind regards,

=====

Referee #3:

The authors completed a substantial amount of work to greatly strengthened support for the conclusions made and improved the manuscript, which describes an interesting topic. However, a few points remain for consideration:

The authors need to further address the specificity of the DSK2 antibody. The response states that the "DSK2 antibody can detect both DSK2A and DSK2B". If this is the case, then both DSK2 isoforms, represented by two protein bands, should be detected in the Col sample lane on the blot shown. The response also states "the DSK2 antibody detected a lower level of DSK2 signal in dsk2a cr mutant and a complete loss of DSK2 signal in dsk2 cr#1 mutant as compared to Col". It should be noted that the loading control indicates that the dsk2a c and dsk2 cr#1 lanes are under loaded compared to the Col lane, which may account for the lower level of DSK2 detected. Strongly suggest completing additional blots to fully confirm specificity of the antibody.

Figure 2C and D - The results clearly shows that the abundance of LAO1 under -N is considerably lower than that observed under +N at each specific timepoint, suggesting that N status impacts protein abundance. However, the increase in LAO1 abundance over time under +N or -N (e.g. compare +N 24hr to 36hr and -N 24hr to 36hr) is not sufficiently addressed. The authors suggest that the increase maybe due to the finding that LAO1 "protein levels are also regulated by developmental stage,

correlating with N assimilation and true leaf development (see response, and fig EV2)". However, the blot in fig. 2C was done using 7-day old seedlings. Why does fig. 2 suggest increase in LOA1 abundance under -N, but fig. S2 show -N promote degradation.

Appendix Figure S3 - Do these graphs represent the mean of 3 independent experiments as mentioned in the legend of Fig. 3B/3C? Graphs in Appendix Figure S3 do not have error bars. Similar question for Appendix Figure S2C.

Referee #3:

The authors completed a substantial amount of work to greatly strengthened support for the conclusions made and improved the manuscript, which describes an interesting topic.

Author Reply: Thank you for your positive feedback and for recognizing the substantial effort we have put into revising and strengthening our manuscript. We sincerely appreciate your comments and suggestions, which have significantly improved our manuscript.

However, a few points remain for consideration:

The authors need to further address the specificity of the DSK2 antibody. The response states that the "DSK2 antibody can detect both DSK2A and DSK2B". If this is the case, then both DSK2 isoforms, represented by two protein bands, should be detected in the Col sample lane on the blot shown. The response also states "the DSK2 antibody detected a lower level of DSK2 signal in *dsk2a* cr mutant and a complete loss of DSK2 signal in *dsk2* cr#1 mutant as compared to Col". It should be noted that the loading control indicates that the *dsk2a* c and *dsk2* cr#1 lanes are under loaded compared to the Col lane, which may account for the lower level of DSK2 detected. Strongly suggest completing additional blots to fully confirm specificity of the antibody.

Author Reply: Thank you for your comments. Several studies have used DSK2 antibody for western blots, but these have yielded varying results. In Lin et al., 2011 (*The Defective Proteasome but Not Substrate Recognition Function Is Responsible for the Null Phenotypes of the Arabidopsis Proteasome Subunit RPN10*. <https://doi.org/10.1105/tpc.111.086702>), multiple blots using anti-DSK2 antibody were presented. Figure 1A, for instance, shows two bands in the left panel, while the right appears to detect only one band. Similarly, blots in Figure 7A also show a single band.

(Figure 1A, Lin et al., 2011)

(Figure 7A, Lin et al., 2011)

The same DSK2 antibody was employed in another study (Nolan et al., 2017. *Selective Autophagy of BES1 Mediated by DSK2 Balances Plant Growth and Survival*), which detected only one band in Col, as seen in Figure 4F. Overall, while this DSK2 antibody was intended to detect both isoforms, it predominantly detects a single band in Col protein samples. The predicted molecular weights for DSK2A and DSK2B are 57.30 KD and 58.07 KD, respectively, making it challenging to distinguish the two isoforms based solely on size differences using standard PAGE gels.

Figure 4F, Nolan et al, 2017

Regarding the DSK2 antibody used in this study, we believe it can detect both isoforms based on the high sequence similarity between DSK2A and DSK2B. We acknowledge that the protein samples might have been unevenly loaded in the western blot results previously shown. To address this, we re-grew Col, *dsk2a cr*, and *dsk2cr#1*, and re-extracted total proteins for western blot analysis using our DSK2 antibody. As shown below, the results consistently show a reduced DSK2 signal in the *dsk2a cr* mutant and a complete loss of DSK2 signal in the *dsk2 cr#1* mutant compared to Col, strongly supporting our previous conclusions.

Figure 2C and D - The results clearly shows that the abundance of LAO1 under -N is considerably lower than that observed under +N at each specific timepoint, suggesting that N status impacts protein abundance. However, the increase in LAO1 abundance over time under +N or -N (e.g. compare +N 24hr to 36hr and -N 24hr to 36hr) is not sufficiently addressed. The authors suggest that the increase maybe due to the finding that LAO1 "protein levels are also regulated by developmental stage, correlating with N assimilation and true leaf development (see response, and fig EV2)". However, the blot in fig. 2C was done using 7-day old seedlings. Why does fig. 2 suggest increase in LOA1 abundance under -N, but fig. S2 show -N promote degradation.

Author Reply: We sincerely appreciate your continued attention to the regulation of LAO1 protein

abundance under nitrogen starvation conditions. We acknowledge your agreement that the results in Figure 2C and D demonstrate the impact of N status on LAO1 protein abundance. To address your concerns more clearly, we would like to provide the following detailed explanation.

Methodologically, Figure 2C and Appendix Fig. S2 are different. Figure 2C shows the impact of N starvation on LAO1 protein abundance at various developmental stages. Seven-day-old seedlings were subjected to nitrogen starvation treatment, yet developmental progression clearly influences LAO1 abundance in this context. To accurately assess the impact of nitrogen status, -N conditions must be compared with +N at each time point to rule out the influence of developmental progression. Conversely, Appendix Fig.S2 employed CHX to block protein synthesis and monitor protein stability within a short time window, thereby eliminating the interference of developmental progression. The CHX chase assays provide a direct measure of protein stability and degradation rates. Despite the increase from 24 to 36 hrs under -N conditions, LAO1 protein levels remain consistently lower under -N than under +N at each time point in Figure 2C, and LAO1 degrades more rapidly under -N conditions in Fig.S2, reinforcing the conclusion that -N destabilizes LAO1. To avoid potential misunderstandings, we have added a sentence (Line 202-206) to remind readers to pay attention to the conditions used in Figure 2C and to compare -N data with +N at each time point.

Appendix Figure S3 - Do these graphs represent the mean of 3 independent experiments as mentioned in the legend of Fig. 3B/3C? Graphs in Appendix Figure S3 do not have error bars. Similar question for Appendix Figure S2C.

Author Reply: We sincerely appreciate your attention to the details in our manuscript and thank you for raising this important point. We apologize for the lack of clarity in the presentation of the data in Appendix Figures S2C and S3. The values in these graphs represent the mean of three independent experiments. We omitted the error bars in these graphs because adding them, along with individual data points as requested by journal policy, would have cluttered the graphs and reduced readability. However, we agree this information should be clearly provided. In the revised manuscript, we have included the mean \pm SD values and have specified the number of samples analyzed in the figure legends.

Dr. Jie Dong
Zhejiang University
Institute of crop science
China

Dear Dr. Dong,

I am very pleased to accept your manuscript for publication in the next available issue of EMBO reports. Thank you for your contribution to our journal.

Yours sincerely,
